# Suspicion-Agent: Playing Imperfect Information Games with Theory of Mind Aware GPT-4

## Abstract

Unlike perfect information games, where all elements are known to every player, imperfect information games emulate the real-world complexities of decision-making under uncertain or incomplete information. GPT-4, the recent breakthrough in large language models (LLMs) trained on massive passive data, is notable for its knowledge retrieval and reasoning abilities. This paper delves into the applicability of GPT-4's learned knowledge for imperfect information games. To achieve this, we introduce **Suspicion-Agent**, an innovative agent that leverages GPT-4's capabilities for imperfect information games. With proper prompt engineering to achieve different functions, Suspicion-Agent based on GPT-4 demonstrates remarkable adaptability across a range of imperfect information card games. Importantly, GPT-4 displays a strong high-order theory of mind (ToM) capacity, meaning it can understand others and intentionally impact others' behavior. Leveraging this, we design a planning strategy that enables GPT-4 to competently play against different opponents, adapting its gameplay style as needed, while requiring only the game rules and descriptions of observations as input. In the experiments, we qualitatively showcase the capabilities of Suspicion-Agent across three different imperfect information games and then quantitatively evaluate it in Leduc Hold'em. As an exploration study, we show that Suspicion-Agent can potentially outperform traditional algorithms without any specialized training or examples, but still cannot beat Nash-Equilibrium algorithms. In order to encourage and foster deeper insights within the community, we make our game-related data publicly available.

## 1 Introduction

Recently, large language models (LLMs) (Brown et al., 2020b; Chowdhery et al., 2022; Touvron et al., 2023), which are trained extensively on text corpora and code datasets and aligned with instructions (Ouyang et al., 2022; Wei et al., 2021; Longpre et al., 2023), have demonstrated remarkable knowledge retrieval and reasoning capabilities (Kojima et al., 2022; Wei et al., 2022b;a) on natural language benchmarks and exams (Hendrycks et al., 2020; Cobbe et al., 2021). Given few-shot examples or specific instructions as prompts, these models, especially GPT-4 (OpenAI, 2023), can understand human intentions and make informed decisions in open-ended scenarios, and tackle intricate tasks by gathering observations and utilizing the learned prior knowledge, such as Voyager (Wang et al., 2023a), ReAct (Yao et al., 2022) and SwiftSage (Lin et al., 2023).

However, most of these methods typically assume that the agent has access to all relevant information, an assumption that is often unrealistic in real-world settings. Take diplomacy (Team et al., 2022; Gray et al., 2020) as an example: representatives must discern the veiled intentions of other countries based on incomplete information and decide accordingly to maximize benefits for their own nation. This challenge is not unique to diplomacy but extends to other domains as well, such as poker (Moravčík et al., 2017; Brown & Sandholm, 2018) and economic simulations (Holmström & Myerson, 1983; Harsanyi, 1968). The inherent unpredictability in these games makes it impractical for a learned agent to adopt a single, optimal strategy for every scenario (Brown et al., 2019). This necessitates predictive capabilities for handling incomplete information, along with a theory of mind (ToM) ability (Frith & Frith, 2005) to comprehend decisions from others' perspectives. Such complexities, both strategic and cognitive, represent ongoing challenges in the field of AI research.

Furthermore, recent advancements in imperfect information games, such as ReBel (Brown et al., 2020a), DeepStack (Moravčík et al., 2017), and Libratus (Brown & Sandholm, 2018), typically start training from scratch, and thus they normally need millions of data to understand the game rules

and learn the adequate strategies for each new game. Such a high sample complexity hampers their ability to generalize across different games and poses challenges when applying them into complex and open-ended imperfect information scenarios. By contrast, as alluded to previously, LLMs have undergone training on massive passive datasets. This leads to an intriguing proposition: Can we harness pre-trained LLMs' knowledge and reasoning capabilities to navigate imperfect information games without additional training or data?

To achieve this, we propose **Suspicion-Agent**, an innovative autonomous agent based on GPT-4. This agent harnesses its extensive prior knowledge and cognitive adaptability to effective strategies against a range of adversaries without any specialized training. Concretely, we first decompose the process of solving such games into multiple sub-modules like observation interpreter and planning module to understand the game rules and game states (as Figure 1 shows) so that GPT-4 can make decisions accordingly. Each module employs different prompts to enable GPT-4 to fulfill specific functions. However, unlike perfect information games, planning strategies in imperfect information games can have varying effectiveness depending on the opponent's behavior (Brown et al., 2020a; Moravčík et al., 2017; Brown & Sandholm, 2018). To tackle these challenges, we introduce a theory of mind (ToM) aware planning approach that leverages the higher-order ToM capability (Frith & Frith, 2005) present in LLMs. Specifically, the model utilizes its understanding of human cognition to predict opponents' thought processes, susceptibilities, and actions. This aligns with the idea that individuals use their own minds as models to understand and affect others (Montes et al., 2022). For instance, the model might consider, *"If I execute Plan 1, how would this influence my opponent's beliefs about my cards, and what actions might they take based on their behavioral patterns?"*

Concretely, given the gameplay history as the input, we find that GPT-4 can identify an opponent's strategic tendencies and analyze how our actions influence the opponent's behavior, *e.g.* if Suspicion-Agent identifies a weak hand held by the opponent, coupled with the cautious strategy, it might strategically raise the bet to encourage the opponent to fold, even when Suspicion-Agent itself holds a similarly weak hand (as illustrated in Figure 9 and K). Remarkably, by using some simple prompts, *e.g.*, GPT-4 can even self-examine its behavior through the lens of the opponent (Refer to G). Leveraging its ToM capabilities, GPT-4 can predict and even influence an opponent's actions effectively (Röska-Hardy, 2008). Integrating these simulated actions into our planning module can mitigate the information asymmetry inherent in imperfect information games and more accurately assess the effectiveness of various strategies. As a result, our Suspicion-Agent can adjust its strategy to play effectively against a range of opponents, as shown in Section 4.1. In the experiments, we first conduct a qualitative assessment of Suspicion-Agent's in 3 two-player imperfect information games, aiming to showcase the generalization capabilities of our method. Subsequently, we perform a quantitative analysis in Leduc Hold'em (Southey et al., 2012). The results reveal that Suspicion-Agent exhibits varying behaviors when interacting with previous works such as CFR+ (Tammelin, 2014) and NFSP (Heinrich & Silver, 2016) while outperforming some of them in terms of overall performance. In summary, our contributions are as follows:

1. We introduce Suspicion-Agent, **the first agent framework designed to empower GPT-4 with theory of mind (ToM) ability to compete in various imperfect information games by understanding game rules and observational data without requiring any specialized training or examples.** By incorporating the ToM capability into the planning process, Suspicion-Agent captures the inherent uncertainty of opponent behavior in our strategic deliberations. This enables Suspicion-Agent to adapt its tactics dynamically when facing opponents with differing behavioral patterns.

2. We are the first to demonstrate that an agent based on GPT-4 can potentially outperform traditional algorithms in imperfect-information games, such as Leduc Hold'em (Southey et al., 2012), when compared to established learning-based methods like NFSP (Heinrich & Silver, 2016) and DMC (Zha et al., 2021), but still unperformed Nash-Equilibrium algorithms like CFR+ (Tammelin, 2014) which may inspire more subsequent use of LLMs in imperfect-information games.

3. We make all interaction data between Suspicion-Agent and traditional algorithms for imperfect-information games in Leduc Hold'em publicly available. This will enable the research community to scrutinize the capabilities of GPT-4-based agents and inspire further work, particularly in fine-tuning smaller language models.

## 2 PROBLEM DEFINITION

**Two-Player Imperfect Information Game** In this paper, we propose to employ LLMs to play imperfect information games. As a preliminary exploration, we concentrate primarily on two-player

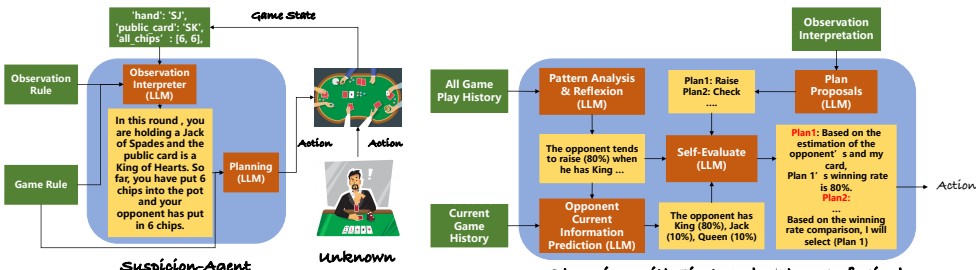

Figure 1: **Left Figure**. The illustration of the Suspicion-Agent which trains the ego policy by pairing it with the copied partner policy. **Right Figure**. The illustration about the first-order ToM planning method, where the texts in yellow blocks are outputs, and green blocks are inputs.

imperfect information games, such as Leduc Hold'em (Southey et al., 2012), which involves two players, denoted by $\mathcal{N} = \{1, 2\}$, who share the same action space, $\mathcal{A}$. Let $a_1 \in \mathcal{A}$ and $a_2 \in \mathcal{A}$ represent the actions chosen by player 1 and player 2, respectively. Each player has access to two types of observations: a **private observation**, denoted as $S_{\text{pri}(i)}$ where $i \in \mathcal{N}$ is the player index, and a **public observation**, shared among both players, denoted as $S_{\text{pub}}$.

As the game progresses in discrete timesteps indexed by $j$, each player $i$ observes a history $h$ of the game. This history comprises the series of public and private observations and actions up to timestep $j - 1$ and the result of game $r^j$, formally given as $h = (S_{\text{pub}}^0, S_{\text{pri}(i)}^0, a_i^0, a_{\neg i}^0, r^0 \ldots, S_{\text{pub}}^{j-1}, S_{\text{pri}(i)}^{j-1}, a_i^{j-1}, a_{\neg i}^{j-1}, r^{j-1})$. Simultaneously, players receive the current private and public observations, $S_{\text{pri}(i)}^j$ and $S_{\text{pub}}^j$, and select the next action $a_i^j$ according to a policy $\pi_i$. All game histories are constructed as a dataset $D$, denoted as $D = (h_1, h_2, \ldots, h_M)$, where $M$ indexes individual games. The goal of each player is to select the next action $a_i^j$ with the imperfect observation according to the game rules, aiming for victory over many games. Specifically, the order of players is not fixed and depends on the game rule for each game. For example, the role of the small blind rotates among players in Texas Holed'em, dictating the order of play.

## 3  METHOD

To enable LLMs to play various imperfect information games without specialized training, we break down the overall task into several modules shown in Figure 1, such as the observation interpreter, game pattern analysis, and planning module. In the following sections, we will demonstrate how we craft specific prompts to guide LLMs to use its prior knowledge, reasoning ability, and psychological ability in performing these modular functions and explain how we combine these functions to equip the model with the capability to navigate the intricacies of imperfect information games. All prompts and codes will be made public on our codebase (Please refer to our supplementary material).

### 3.1  GAME RULE & OBSERVATION UNDERSTANDING

While LLMs excel in processing text data, it can be misled in imperfect information games because they normally provide only brief, low-level descriptions. To mitigate this issue, we initially develop structured prompts that assist LLMs in comprehending both the game's rules and its current state. For each type of imperfect information game, one can write a structured rule description as follows:

- **General Rules:** A brief game introduction, the number of rounds, and betting rules;
- **Action Descriptions:** {Description of Action 1}, {Description of Action 2}, ...;
- **Single Win/Loss Rule:** The conditions for winning, losing, or drawing in a single game;
- **Win/Loss Payoff Rule:** The rewards or penalties for winning or losing a single game;
- **Whole Win/Loss Rule:** The number of games and the overall win/loss conditions.

In most imperfect information game environments (Zha et al., 2019), game states are often represented as low-level numerical values, such as one-hot vectors, to facilitate machine learning. Leveraging LLMs, we can convert these low-level game states into natural language text (Wu et al., 2023; Wang et al., 2023a; Guo et al., 2022; Lin et al., 2023), thereby aiding the model's understanding. For each game, it is also essential to define an observation conversion rule. Similar to structuring game rules, we organize the observation conversion rule as follows:

- **Input Explanation:** The type of inputs received, such as dictionaries, lists, or other formats, and describes the number of elements in the game state along with the name of each element;

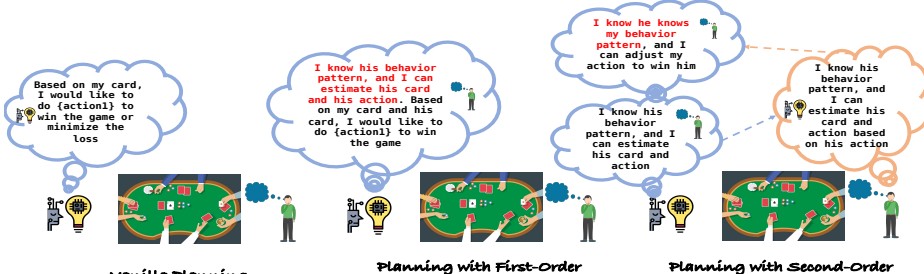

Figure 2: **Left Figure**. The decision-making of the vanilla planning of Suspicion-Agent. **Middle Figure**. The decision-making of the planning with first-order ToM of Suspicion-Agent. **Right Figure**. The decision-making of the planning with second-order ToM of Suspicion-Agent.

- **Element Descriptions:** {Description of Element 1}, {Description of Element 2}, ...;
- **Conversion Tips:** More guidelines for transforming the low-level game states into text.

By leveraging both the game rule and the observation conversion rule, we can efficiently transform low-level game states into readable text, denoted as $Obs_r$. This readable text serves as the input for LLMs. Using the prompts $Prompt_{obs}$, the conditional distribution for each element $Obs_r[i]$ in the generated text can be modeled as: $Obs_r \sim \prod_{i=1}^{M} F_\theta(Obs_r[i]|Prompt_{obs}, Rule, Rule_{obs}, Obs_r[1, \ldots, i-1])$. Here, $F_\theta$ represents the language model parameterized by $\theta$; $M$ is the length of the generated text $Obs_r$. The concrete definition can be found in Appendix A. We name this module an **Observation Interpreter**. This formulation allows for a more understandable interaction with the model in imperfect information games.

## 3.2 Vanilla Planning Module and Reflexion

After understanding the game rules and converting the game states into a readable format, we can craft prompts to guide LLMs in formulating strategies. Inspired by advancements in LLMs-agent and prompt engineering (Ganguli et al., 2023; Wang et al., 2023d; Liu et al., 2023b; Shinn et al., 2023), we introduce a vanilla planning method which features a **Reflexion** module aimed at automatically scrutinizing game history to enable LLMs to learn and improve planning from the experience of the history, as well as a separate **planning** module dedicated to making decisions accordingly.

**Reflexion** The Reflexion module takes as input the history of games played against the current opponent and outputs a Reflexion. In the $j$-th round of the $i$-th game, we gather all prior game histories, denoted as $D^i = (h^1, h^2, \ldots, h^{i-1})$, and prompt LLMs to carry out these analytical functions to get the Reflexion output $O_f^i \sim \prod_{i=1}^{M} F_\theta(O_f[i]|Prompt_{Reflexion}, Rule, D^i, O_f[1, \ldots, i-1])$, i.e., $O_f^i \sim F_\theta^{Reflexion}$, which covers why we won or lost in specific previous games, and suggests how to improve strategies for future games. Importantly, the Reflexion module empowers Suspicion-Agent to enhance its strategies during gameplay, even without previous examples.

**Planning** After obtaining the Reflexion $O_f$, we proceed to use the game rules, the current game history $h^i$, the current readable observation $Obs_r$, and the set of valid actions $\{a\}$ in the current game as inputs. We then prompt LLMs to formulate multiple textual plans based on its understanding of the imperfect information, i.e., $O_{plan} \sim \prod_{i=1}^{M} F_\theta(O_{plan}[i]|Prompt_{plan}, Rule, Obs_r, h^{j-1}, O_f, O_{plan}[1, \ldots, i-1]), O_{plan} \sim F_\theta^{plans}$. Specifically, the vanilla planning method assumes the marginal distribution of the actions of the opponent is uniform, and thus it can be regarded as a special case of planning with the zero-order ToM. In this way, we can further denote $F_\theta^{plans}$ as $F_\theta^{zero-plan}$.

**Evaluator** To assess the likely success of each plan, we introduce an evaluation module. This module takes into account factors such as the game's current state, i.e., readable observation $Obs_r$, the Reflexion $O_{Reflexion}$, the game rule $Rule$ and estimated plans $O_{plan}$ as the input, to estimate the win rates for each of the proposed plans and output the next action by prompting LLMs, i.e., the next action $a_j = F_\theta^{zero-eval}(Obs_r, O_{Reflexion}, Rule, O_{plan})$.

## 3.3 Planning with Theory of Mind (ToM)

However, the vanilla planning method often struggles against the inherent uncertainties that typify imperfect information games, particularly when faced with opponents skilled at exploiting others'

strategies. Inspired by this adaptability, we seek to devise a new planning method that capitalizes on LLMs' ToM capabilities (Frith & Frith, 2005; Kosinski, 2023) to understand the opponent's behavior and thus can adjust the strategy accordingly. In the following sections, we will detail how we employ LLMs to analyze the behavior patterns of other agents and predict their subsequent actions in response to various plans using different orders of ToM (results are shown in Table 3), thereby facilitating more informed decision-making. Note that all sample outputs are given in Section K and G.

**Planning with First-Order ToM Modelling**: In the first-order ToM modeling approach (as Figure 7 shows), Suspicion-Agent goes a step further by inferring the probable hand of the opponent based on their actions to that point, *e.g.*, if the opponent raised, they likely have a strong hand. Consequently, Suspicion-Agent can adapt their strategy to maximize winnings when holding a strong hand and minimize losses with a weak hand. To forecast the opponent's actions, we first introduce a behavior pattern analysis process. In this process, we feed the game history $D^i = (h^1, h^2, \ldots, h^{i-1})$ and the game rules into LLMs, prompting it to analyze the opponent's behavioral pattern. The formulation and the prompts can be expressed as: $O_{bp} \sim \prod_{i=1}^{M} F_\theta(O_{bp}[i]|\text{Prompt}_{\text{pattern}}, \text{Rule}, D^i, O_{bp}[1, \ldots, i-1])$.

**Sample Prompts for First-Order Behaviour Pattern Analysis (Incomplete) :** *From my perspective, please infer several beliefs about the opponent's game pattern/preference for each round when holding different cards and the public card (if have).*

Through this approach, we can deduce the opponent's behavior pattern. Notably, since the input for behavior pattern analysis is the same as that for the Reflexion module, we have integrated them into a single module to reduce inference time, as shown in Figure 1. After identifying the opponent's behavior pattern, LLMs can be prompted to predict the strength of the opponent's current hand or observations in the game. This is expressed as: $O_{\text{card\_pred}} \sim \prod_{i=1}^{M} F_\theta(O_{\text{card\_pred}}[i]|\text{Prompt}_{\text{card\_pred}}, \text{Rule}, h^{j-1}, O_{bp}, \text{Obs}_r^j, O_{\text{card\_pred}}[1, \ldots, i-1])$.

**Sample Prompts for First-Order Cards Prediction (Incomplete) :** *Understanding the game rule, your observation, progress summarization in the current game, the estimated behaviour pattern of the opponent, and your knowledge about the game, please infer the probabilities about the cards of the opponent (number 100% in total) step by step.*

With these predictions, we can further augment the previous **Planning** module and **Evaluator** module with $O_{\text{card\_pred}}$ as the additional input, so that we can further propose better plans considering the opponent's card and estimate the winning rate of each plan, so that we can better make the decision. Because the input of **Planning** module and **Evaluator** module are highly overlapped and our budgets are limited, we combine these two modules together to save the costs:

**Sample Prompts for Planning and Evaluator (Incomplete):** *Make Reasonable Plans: Please plan several strategies according to actions you can play now to win the whole game step by step. Note that you can say something or keep silent to confuse your opponent.*

*Potential opponent's actions and Estimate Winning/Lose/Draw Rate: From the perspective of the opponent, please infer what the action opponent with probability would do when the opponent holds different cards based on his behaviour pattern, and then calculate the winning/lose/draw rates when opponent holds different cards step by step. Output in a tree structure:*

The sample outputs are shown in Figure 8 and G.1.

**Planning with Second-Order ToM Modelling**: However, elite players in imperfect information games like poker are also adept at dynamically adjusting their strategies, and they may employ "bluffing" as a tactic, feigning a strong hand when they actually hold a weak one to deceive their opponent. Relying solely on a first-order ToM in such situations could lead to incorrect assumptions and potentially costly mistakes. Recognizing this, we introduce a planning method that incorporates a second-order ToM. In this enhanced model, Suspicion-Agent engages in even more intricate reasoning, where Suspicion-Agent not only considers what the opponent might do (as in first-order ToM) but also what the opponent believes Suspicion-Agent will do as Figure 7 shows. This level of strategic thinking allows Suspicion-Agent to gain an advantage in situations involving tactics like bluffing.

To implement this, Suspicion-Agent needs to not only just consider the current state from its own perspective, but also be capable of role-switching to think his own observation from the opponent's viewpoint. In traditional methods (De Weerd et al., 2013; Tatarchenko et al., 2016), they need to iteratively call the first-order ToM function to estimate the action of the opponent. However, we surprisingly find that we can just add the prompts like below, and get the outputs in Sec G.2.

**Sample Prompts for Second-Order Behaviour Pattern Analysis (Incomplete):** *From my perspective, please infer under what circumstances is the opponent likely to be influenced by my actions? Additionally, in what situations would the opponent make decisions based solely on their own hand?*

*From the perspective of the opponent (he cannot observe my card but only action), please infer several beliefs about my game pattern/preference when holding different cards.*

With this, LLMs are able to automatically generate insights into whether the opponent's behavior is likely to be reactive to Suspicion-Agent's actions, or if they are likely to act independently. Then, we can directly reuse the prompts of the first-order Tom to predict the opponent's cards based on the behavior pattern estimated from the second-order ToM, and we can get sample results in Figure 9 and Section G.2. In this way, we can utilize the Planning with second-order ToM to make decisions and adapt the strategies accordingly. The concrete algorithms are given in Section H. Without mentioning otherwise, we use the second-order ToM and GPT-4-0613 by default.

## 4 EXPERIMENTS

We conduct experiments to answer the following questions:

- Can Suspicion-Agent achieve comparable performance with traditional imperfect information algorithms without any specialized training? (Section 4.1)
- Can Suspicion-Agent adapt its strategies when playing with different opponents? (Section 4.1)
- Can Suspicion-Agent play different imperfect information games without any specialized training? (Section K.1)
- How different orders of ToM improve the performance of Suspicion-Agent? (Section 4.3 and I)

### 4.1 QUANTITATIVE EVALUATION

**Environments** To quantitatively assess the performance of LLMs in imperfect information games, we chose the RLCard environment (Zha et al., 2019). Due to budget limits, our quantitative evaluation focuses on Leduc Hold'em [1], a simplified version of Limit Texas Hold'em. The game rules of Leduc Hold'em can be found in Appendix B. Following (Southey et al., 2012), we also add the opponent's observation into the single game history $h$ after the end of each game, which also conforms with the real-world experience, but we also perform the ablation study about it in Section 4.3 and J.

**Competing Methods** We have selected a range of methods commonly used in decision-making, such as NFSP (Heinrich & Silver, 2016), DQN (Mnih et al., 2015), DMC (Deep Monte Carlo Search for imperfect information games) (Zha et al., 2021) and CFR+ (Zinkevich et al., 2007). Among these, NFSP and DMC are specifically designed for imperfect information games and are based on self-play, while CFR+ is grounded in game theory. These algorithms typically show different strategies in the imperfect information games, allowing us to evaluate the adaptability of each method. Note that, our Suspicion-Agent does not have any specialized training when compared with these methods.

**Evaluation Methods** To ensure the robustness of our evaluation metrics, we meticulously designed a dual-method evaluation framework aimed at mitigating the randomness intrinsic to imperfect information games. **(1) Variable Random Seeds:** Suspicion-Agent play against different baselines for 100 games utilizing varying random seeds for each game. This tactic is intended to dampen the stochastic variability introduced by the random seed settings. The results are shown in Table 1. **(2) Same Cards with Exchange Position:** We ran a series of 50 games with a fixed random seed, thereby keeping the sequence of cards constant across these games. Suspicion-Agent initially played at position 0 for the first 50 games, then we rerun the 50 games but switched the position of Suspicion-Agent and the baseline model. In this way, Suspicion-Agent and the baseline should have the same card strength over 100 games, and thus we can better evaluate the performance of each. The results of these experiments are presented in Table 2.

**Results Analysis** (1) **Suspicion-Agent outperforms most baselines:** As illustrated in Table 1, it is evident that our GPT-4-based Suspicion-Agent outperforms most other algorithms specifically trained on Leduc Hold'em environments except CFR+. Notably, it not only defeats most of these methods but also secures the highest average chip count in the comparisons. Our approach surpasses the second-best method by an impressive margin of approximately 25%. Even if the current design

---

[1] https://rlcard.org/games.html

cannot achieve Nash-Equilibrium, these findings compellingly showcase the potential of employing large language models in the realm of imperfect information games, as well as affirm the effectiveness of our proposed framework. (2) **The gap between GPT-3.5 and GPT-4 is large:** While GPT-4 delivers performance that either matches or outperforms other baselines, agents using GPT-3.5 experience a significant drop in performance. Specifically, the winning probability for agents built on GPT-3.5 stands at just 50%, as opposed to 100% for GPT-4-based agents. Additionally, the average chip payoff for GPT-3.5 agents is negative, underlining the stark performance disparity between the two versions of the language model. The further reason analysis can be found in Appendix C. (3) **Suspicion-Agent outperforms baselines in both positions:** Utilizing identical card sequences for both positions, Suspicion-Agent exhibits a consistent winning pattern against various baselines, as evidenced in Table 2. This robust performance serves as compelling evidence to substantiate the claim that Suspicion-Agent outperforms the baseline models when card strength is held constant.

Table 1: The comparison results of Suspicion-Agent when playing with different algorithms trained on Leduc Hold'em environments. The results are the win/lose chips after 100 games with different seeds, and the number of win/lose chips ranges from 1 to 14.

|  | Opponent Model | | | | | | |
|---|---|---|---|---|---|---|---|
|  | NFSP | DQN | DMC | CFR+ | Ours (GPT-3.5) | Ours (GPT-4) | Avg. |
| NFSP | - | -33 | -22 | -21 | -3 | -142 | -61.25 |
| DQN | +33 | - | -55 | -8 | +200 | -44 | +22.8 |
| DMC | +22 | +55 | - | -15 | -49 | -24 | +4 |
| CFR+ | +21 | +8 | +15 | - | +126 | +22 | +38.4 |
| Ours (GPT-3.5) | +3 | -200 | +49 | -126 | - | - | -55 |
| Ours (GPT-4) | **+142** | +45 | +24 | -22 | - | - | **+47.25** |

Table 2: The comparison results of Suspicion-Agent when playing with CFR+ and DMC trained in Leduc Hold'em environments. These results are quantified over 50 games, and pos denote the position of the opponent model. For example, CFR+ (pos 0) denotes the opponent model is in the position 0, and the model is located in the position 1.

|  | Opponent Model | | | | | | |
|---|---|---|---|---|---|---|---|
|  | CFR+ (pos 0) | CFR+ (pos 1) | DMC (pos 0) | DMC (pos 1) | Ours (pos 0) | Ours (pos 1) | Avg. |
| DMC | -21 | -6 | -10 | +10 | -36 | -4 | -11.17 |
| CFR+ | +35 | -35 | +6 | +21 | +16 | -1 | +7 |
| Ours | +1 | **-16** | +11 | **+36** | - | - | **+8** |

**Behaviour Pattern Analysis** We illustrate the action percentages of Suspicion-Agent and baselines in Figure 3. We can observe that (1) **Suspicion-Agent vs CFR+:** CFR+ algorithm (Zinkevich et al., 2007) exhibits a mixed strategy, characterized by a conservative approach where it tends to fold in the face of a weak hand, especially when this hand does not align well with the public cards. Our agent, recognizing this pattern, strategically opts to raise more frequently. This tactic effectively applies pressure on the CFR+ algorithm, increasing the likelihood of it folding under uncertain conditions. However, Over-reliance on bluffing can lead to significant losses, especially when facing opponents capable of recognizing and exploiting this pattern. Additionally, it's noteworthy that CFR+ itself is not averse to bluffing. Although it typically plays conservatively, CFR+ does occasionally employ bluffing tactics, which can result in losses for our agent. (2) **Suspicion-Agent vs DMC:** DMC algorithm (Zha et al., 2021) based on the search algorithm DMC employs a more diversified strategy that includes bluffing. It often raises both when it has the weakest and the strongest hands. In response, Suspicion-Agent adapts by raising less frequently and opting to call or fold more often based on its own hand and the observed behavior of DMC. (3) **Suspicion-Agent vs DQN:** DQN appears to have a more aggressive stance, almost always raising with strong or mid-level hands and never folding. Suspicion-Agent identifies this and, in turn, minimizes its own raises (the lowest percentage among all matchups), opting more often to call or fold based on the public cards and DQN's actions. (4) **Suspicion-Agent vs NFSP:** NFSP exhibits a follow-through strategy, opting to always call and never fold. Suspicion-Agent responds to this by raising less frequently (compared to matches against CFR+) and choosing to call more (compared to matches against CFR+) based on the public card and NFSP's observed actions. The analysis clearly shows that Suspicion-Agent is highly adaptable and capable of exploiting the weaknesses in the strategies employed by various other algorithms. This speaks volumes about the large language model's capability to reason and adapt in imperfect information games.

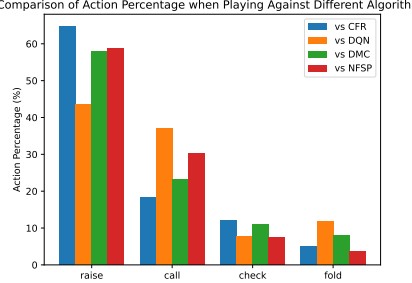
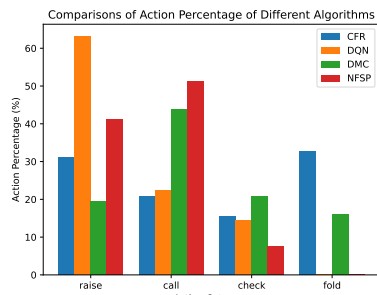

Figure 3: **Left Figure**. The illustration of the action percentage of Suspicion-Agent when playing against different algorithms. **Right Figure**. The illustration of the action percentage of different algorithms when playing against Suspicion-Agent.
.

## 4.2 QUALITATIVE EVALUATION

In the qualitative evaluation, we assess Suspicion-Agent on three imperfect information games: Coup, Texas Hold'em Limit, and Leduc Hold'em (Southey et al., 2012). For each game, we provide only the rules and observation rules as described in Section 3.1. Importantly, Suspicion-Agent is able to play these games without any additional training or sampling. Qualitative examples from these games are presented in the subsequent sections.

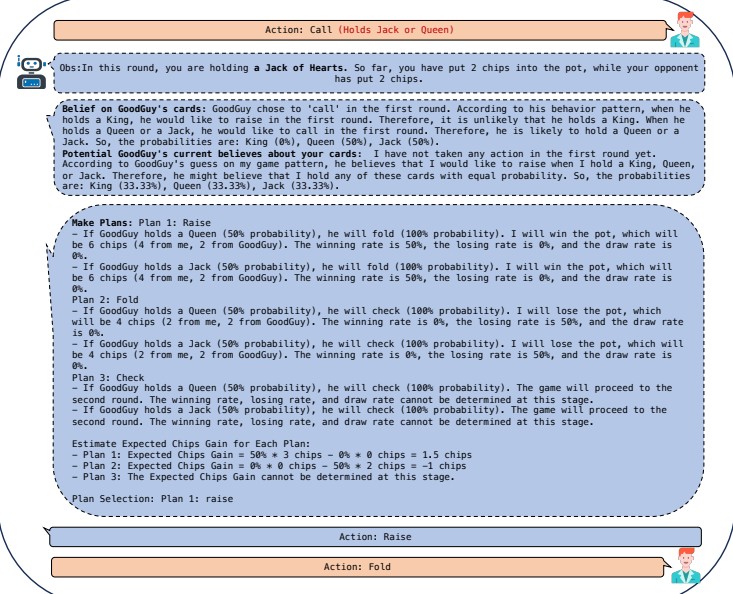

Figure 4: The qualitative sample of planning with second-order ToM Suspicion-Agent about **Strategic Bluffing** on Leduc Hold'em. More samples are given in Appendix K.

**Leduc Hold'em** We present qualitative samples showcasing Suspicion-Agent's behaviour under different strategies: Vanilla Planning, Planning with First-Order ToM, and Planning with Second-Order ToM in Leduc Hold'em. These samples can be viewed in Figure 6, 7, 8, and 9, respectively, and the concrete analysis is given in Appendix K.1.

**Game Coup and Texas Hold'em Limit** As illustrated in Figure 12 and 11 in Appendix, when provided solely with the rules and observation guidelines of Texas Hold'em Limit and Coup, and keeping the agent prompts consistent, Suspicion-Agentis still adept at discerning the opponent's game patterns. It analyzes the strength of its hand and subsequently makes informed decisions to accumulate chips. This is the strong evidence to demonstrate the generalization ability of Suspicion-Agent based on GPT-4 on different imperfect information games, and thus outperforming the algorithms need to be re-trained for every new imperfect information game. Specifically, In the game of Coup, without any prior training, **Suspicion-Agent** rapidly discerns which character the opponent lacks. It then strategically bluffs as that character to block the opponent's actions. This ability to bluff successfully gives **Suspicion-Agent** a consistent advantage throughout multiple rounds.

## 4.3 Ablation Study and Component Analysis

Table 3: The comparison results of Suspicion-Agent when playing with different levels of ToM against CFR+ on Leduc Hold'em environments. The results are quantified after 100 games.

|  | Opponent Model | |
|---|---|---|
|  | CFR+ | DMC |
| Ours (vanilla plan) | -89 | -72 |
| Ours (w/ First ToM) | -67 | -26 |
| Ours (w/ Second ToM) | -22 | +24 |

Table 4: The comparison results indicate the impact of including opponent observations in the game history in Leduc Hold'em environments. These results are quantified over 50 games, and pos denote the position of the opponent model.

|  | Opponent Model | | | |
|---|---|---|---|---|
|  | CFR+ (pos 1) | DMC (pos 1) | Avg. | Win Probability |
| DMC | -6 | +10 | +2 | 50% |
| CFR+ | -35 | +21 | -14 | 50% |
| Ours (w/o hind_obs) | -45 | +18 | +17 | **100%** |
| Ours (w/ hind_obs) | -16 | **+36** | **+36.5** | **100%** |

**Ablation Study on Orders of ToM** In order to explore how different orders of ToM-aware planning methods affect the behavior of the large language model. We perform the experiments and comparisons on Leduc Hold'em and play against CFR+. We illustrate the action percentages of Suspicion-Agent with planning with different levels of ToM in Figure 5 and the chips gain results in Table 3. We can observe that (1) **Vanilla Planning:** Based on **Reflexion** module, vanilla planning tends to call and check more (the highest call and check percentage when playing against both CFR+ and DMC) during the game, which cannot push the pressure to make the opponent fold and result in many unnecessary losses. In this way, vanilla planning has the lowest chip gain as Table 3 shows. (2) **Planning with First-Order ToM:** Utilizing First-Order ToM, Suspicion-Agent is capable of making decisions based on its own and estimates of the opponent's card strength. As a result, it will raise more than vanilla planning but it tends to fold more frequently than other strategies, aiming to minimize unnecessary losses. However, this cautious approach can be exploited by savvy opponent models. For example, DMC often raises when holding the weakest hand, and CFR+ may occasionally raise even with a mid-level hand to exert pressure on Suspicion-Agent. In these instances, Suspicion-Agent's tendency to fold can lead to losses. (3) **Planning with Second-Order ToM:** In contrast, Suspicion-Agent excels at identifying and capitalizing on the behavioural patterns of opponent models. Specifically, when DMC checks—suggesting its hand doesn't align with the public cards—Suspicion-Agent will raise as a bluff to induce folds from the opponents. As a result, Suspicion-Agent exhibits the highest raise rate among the three planning methods evaluated. This aggressive strategy allows Suspicion-Agent to accumulate more chips even when holding a weak hand, thereby maximizing its chip gains.

**Ablation Study on the Effect of Hindsight Observation** Following (Southey et al., 2012), we assume that Suspicion-Agent has access to observations of the opponent after the end of each game, *i.e.*, Hindsight Observation. To assess the impact of it, we conduct an ablation study in which hindsight observations are not incorporated into the current game. Without hindsight observations, we augment the **Reflexion** module with additional prompts to enable it to infer the opponent's cards based on game outcomes and Suspicion-Agent's own observations. As demonstrated in Table 5 and 4, Suspicion-Agent retains its performance advantage over the baseline methods without the benefit of hindsight observations. Specifically, we observe that Suspicion-Agent adopts a more conservative strategy under the increased uncertainty that comes without hindsight observations. This leads to reduced bluffing, resulting in fewer gains when playing against CFR+. However, it also minimizes the risk of over-bluffing when facing DMC, thus yielding higher chip earnings.

## 5 Conclusion

In this paper, we introduce Suspicion-Agent, the first prompting system designed to enable large language models to engage in various imperfect information games using only the game rules and observations for interpretation. By incorporating first-order ToM and second-order ToM capabilities, we show that a GPT-4-based Suspicion-Agent can outperform traditional algorithms such as NFSP, even without specialized training or examples. Additionally, we identify and discuss the current limitations of utilizing LLMs in the context of imperfect information games. We make all our code and interactive data publicly available to the research community. This will help in better understanding the capabilities of large language models, particularly GPT-4, and we hope our data will encourage the development of more efficient models for imperfect information games. In addition, we also present the limitations of Suspicion-Agent in Appendix E.

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

## A  BACKGROUND

**Prompting in LLMs**   Recent research has shown that LLMs, notably GPT-4, can function as universal approximators (Schuurmans, 2023; Kim et al., 2023). Given well-crafted prompts, these models can autonomously perform a broad range of tasks using autoregressive text generation (Schuurmans, 2023; Kim et al., 2023). To formalize this, let $F_\theta$ denote a pre-trained LLM characterized by the parameter $\theta$. Consider $L = (L[0], L[1], \ldots, L[M])$ as a sequence of language tokens, where each $L[i]$ is a distinct token. The distribution $F_\theta(L)$ can then be expressed as: $F_\theta(L) = \prod_{i=1}^{M} F_\theta(L[i]|L[0, \ldots, i-1])$. By framing the input to the language model as specific task instructions or as few-shot input-output examples (Kojima et al., 2022; Wei et al., 2022b), it becomes possible to guide $F_\theta$ to generate meaningful output for various tasks. For convenience, let's denote the prompt as $P = (P[0], P[1], \ldots, P[M])$. The output from the LLM, represented by $Y$, can then be defined by the following probability distribution: $p_Y = \prod_{i=1}^{M} F_\theta(Y[i]|P, Y[1, \ldots, i-1])$, where $M$ is the total number of output tokens. For convenience, the conditional function $F_\theta(Y[i]|P, Y[1, \ldots, i-1])$ can be abbreviated as $F_\theta^P$, indicating that it is the function specific to the given prompt $P$.

## B  GAME RULES OF LEDUC HOLD'EM

In Leduc Hold'em, the deck consists only of two Jacks, two Queens, and two Kings. Each player is dealt one of these as a hole card, and a single public card is revealed. The winner is determined by matching the rank of the public card with their hole card; if no such match exists, the player with the higher-ranked hole card wins. Note that the big blind in our setting is 2, and the payoff of a single game ranges from 1 to 14. Following (Southey et al., 2012), we also add the opponent's observation into the single game history $h$ after the end of each game, which also conforms with the real-world experience, but we also perform the ablation study about it in Section 4.3 and J.

## C  GAP BETWEEN GPT-3.5 AND GPT-4

Upon closer examination, we've identified several areas where GPT-3.5 falls short: (a) Instruction Comprehension: GPT-3.5 struggles with understanding instructions as effectively as GPT-4 does. (b) Long Prompt Handling: The performance of GPT-3.5 significantly deteriorates in scenarios involving lengthy prompts. (c) Reasoning and ToM: Even when GPT-3.5 successfully comprehends the instructions, its reasoning and ToM capabilities are markedly inferior to those of GPT-4, resulting in less reasonable outcomes.

## D  RELATED WORKS

**Imperfect Information Game** Imperfect information games, exemplified by poker (Brown & Sandholm, 2018; 2019; Moravčík et al., 2017; Southey et al., 2012), have emerged as captivating subjects of research due to the inherent absence of complete information concerning player states and potential actions (Frank & Basin, 2001). In contrast to perfect information games, these settings allow players to employ strategies that encompass elements of deception and exploration, often leaning towards stochastic approaches over deterministic ones (Montes et al., 2022; Kreps & Wilson, 1982). Previous investigations into imperfect information games have explored a plethora of dimensions, including principles from game theory (Lu et al., 2023b) such as CFR (Zinkevich et al., 2007), CFR+ (Tammelin, 2014) and its optimistic variants (Farina et al., 2019b;a) along with other new versions such as discounted CFR (Brown & Sandholm, 2019) and predictive CFR (Jiang et al., 2022) , techniques from reinforcement learning (Ouyang & Zhou, 2023; Roughgarden, 2016), the integration of deep reinforcement learning (Brown et al., 2020a) such as some decision-making methods already used in card games: NFSP (Heinrich & Silver, 2016), DQN (Mnih et al., 2015), and DMC (Zha et al., 2021), strategies based on observation (Chatterjee et al., 2007), considerations for limited lookahead (Kroer & Sandholm, 2020), and methodologies for abstracting away the intricacies of imperfect information (Sokota et al., 2023). Nonetheless, these approaches often demand extensive computational training and entail the collection of copious behavioral data during gameplay. Recent studies have also delved into the application of LLMs within the context of imperfect information

games (Gupta, 2023). LLMs have not only demonstrated strong performance in decoding and predicting player behaviors (Akata et al., 2023) but have also excelled in simplifying natural language interactions within these gaming environments (Xu et al., 2023). Notably, the zero-shot capabilities of LLMs obviate the necessity for exhaustive pre-training or the accumulation of action data, distinguishing LLM-based methods from the traditional techniques mentioned aforementioned.

**Reasoning and Planning of LLMs** LLMs have recently shown remarkable prowess in reasoning and planning across various downstream tasks (Brown et al., 2020b; Chowdhery et al., 2022; Touvron et al., 2023). They exhibit the ability to employ evidence, arguments, and logic to draw conclusions or make informed judgments (Huang & Chang, 2022). The introduction of the Chain-of-Thought (CoT) approach, which prompts LLMs to generate intermediate reasoning steps, has led to enhanced performance in arithmetic, commonsense, and symbolic reasoning tasks (Wei et al., 2022b). Furthermore, the zero-shot capability of LLMs has proven its potency by simply incorporating a straightforward prompt phrase (Kojima et al., 2022). Subsequently, the Tree-of-Thought (ToT) framework was proposed, enabling exploration over coherent units of text that serve as intermediary steps toward problem-solving, thereby generalizing the popular CoT approach to prompt language models (Yao et al., 2023). Additionally, the Algorithm of Thought (AoT) leverages the inherent recurrence dynamics of LLMs, expanding their idea exploration with minimal queries (Sel et al., 2023). Drawing inspiration from recent developments such as Voyager (Wang et al., 2023a), BabyAGI (Nakajima, 2023), ReAct (Yao et al., 2022), SwiftSage (Lin et al., 2023), Auto-GPT (Richards et al., 2023) and Agent-GPT (Reworkd, 2023), we posit that the reasoning and planning capabilities of LLMs could prove invaluable in supporting agents in imperfect information games, utilizing only the game rules and observations for interpretation. More specifically, when provided with well-crafted prompts, LLM-based agents can autonomously generate a wide array of text, which can be harnessed to facilitate reflection and planning in the context of imperfect information games (Schuurmans, 2023; Kim et al., 2023). Nevertheless, our paper focuses on integrating theory-of-mind (ToM) capacities into the planning process, whereas others do not use them.

**Theory of Mind (ToM)** In the domain of imperfect information games, classical methodologies often draw from game theory. However, these approaches recognize that human decision-making follows a "cognitive hierarchy" of strategies rather than strictly adhering to the hyper-rational Nash equilibrium solution concept (Wunder et al., 2011). The concept of Theory of Mind (ToM) is crucial in understanding human cognitive abilities. ToM involves comprehending and predicting the behaviors of oneself and others by attributing internal mental states like beliefs, knowledge, desires, and intentions (Premack & Woodruff, 1978; Frith & Frith, 2005; Davies, 1994; Nichols & Stich, 2003; Hurley, 2008). In the context of imperfect information games, ToM has been employed to anticipate opponents' actions and observations, enhancing decision-making effectiveness (De Weerd et al., 2013). Level-k thinking theory within cognitive hierarchy theory posits that players in strategic games rely on predictions of other players' likely actions, and it categorizes these players based on the depth of their strategic thinking, a dimension potentially intertwined with ToM (Crawford, 2018). Within the framework of simulating ToM, a player adopts the perspective of their opponent to infer their own potential actions in similar situations. Moreover, ToM can employ recursively nested beliefs through higher-order ToM (Frith & Frith, 2005), enabling not only the assessment of the counterpart's thoughts but also self-reflection on observations and how one's actions may influence the counterpart's future actions. Recent studies have also evaluated ToM capabilities in LLMs (Frith & Frith, 2005; Kosinski, 2023). Nevertheless, there remains a notable gap in the existing literature concerning the integration of ToM within LLMs for imperfect information games.

Limited by our capacity, we are unable to include all relevant literature in our work. If you find any missing relevant references, please feel free to tell us. We appreciate your efforts to improve our paper quality.

## E   LIMITATIONS

**Robustness of Results**   Due to budgetary constraints, our experiments are limited to running 100 games for each comparison with baseline methods. (1) Although this sample size may not be extensive, the superior performance observed over four different baseline algorithms with varying behavioral patterns can still serve as a preliminary demonstration of the cognitive capabilities and potential of large language models like GPT-4 in imperfect information games. (2) Given the same

game sequences, Suspicion-Agent can outperform baselines in both positions. This consistency highlights the adaptability and robustness of Suspicion-Agent even when faced with varied strategies that give the same card strength. Considering the limited budgets and the experimental results we get, it is safe to claim that Suspicion-Agent based on GPT-4 can potentially outperform previous methods designed for imperfect information games.

**Hallucination Problem of Large Language Model**    Hallucination problem (Zhang et al., 2023; McKenna et al., 2023; Bang et al., 2023)—generating outputs that are nonsensical or unfaithful to the provided source content (Ji et al., 2023)—poses a significant challenge in LLMs. In our experiments, we found that when given only simple instructions, LLMs can produce outputs that are either meaningful and rigorous or less rigorous and even invalid. This variability compromises the reliability of LLMs, particularly when they interact with models trained for specialized tasks. In addition, the outputs of LLMs are very sensitive to the prompts. To mitigate this issue, **we developed multiple output templates to improve the quality of the outputs of LLMs**, the effectiveness of which is empirically demonstrated in our main results (these templates will be made publicly available in our code repository). However, further work is needed to better align LLM-generated outputs with given instruction prompts. Enhancing this alignment is a critical area of research for improving the reliability and real-world applicability of LLMs.

**Long Reasoning Problem**    The limitations of LLMs like GPT-4 manifest in two ways when applied to complex tasks in imperfect information games:

1) Long Prompts Problem: To adapt LLMs for different imperfect information games, it is necessary to input both game rules and game conversion rules. When these are combined with the specialized prompts designed for our Suspicion-Agent, the resulting language model prompts become excessively long. We have observed a rapid decline in the quality of the model's output as the length of these prompts increases. Limited by the budgets, we implement the planning and evaluator model into a single function, which results in a quite long sequence generation, leading to the performance decline to some extent.

2) Complex Reasoning/Calculation Problem: When tasked with conducting intricate calculations—such as computing the average win, lose, or draw rate when an opponent holds different cards—GPT-4 struggles to consistently generate accurate mathematical equations and results.

**Expensive Inference Cost and Slow Inference Time**    As demonstrated in Table 1, only GPT-4 is capable of performing well in the game of Leduc Hold'em. Due to the extensive prompts and inference tokens, the cost per game reaches nearly one dollar. Additionally, the large model size of GPT-4 leads to a longer inference time for Suspicion-Agent, requiring several minutes to complete a single game of Leduc Hold'em. These two limitations underscore the importance of developing a specialized local language model for this task, which also serves as our motivation for releasing all associated data.

**Planning Depth**    In our paper, we only focus on single-step planning, but note that our planning method is orthogonal to recently proposed approaches that leverage large language models for planning with depth, such as Tree-of-Thoughts (Yao et al., 2023), Graph-of-Thoughts (Besta et al., 2023), and Algorithm-of-Thoughts (Sel et al., 2023). While these approaches offer promising directions for future research, they come with high computational costs, and thus we do not incorporate them into our current methods.

**More Language Model Evaluation**    In this paper, the evaluation is confined to the performance of GPT-3.5 and GPT-4 on imperfect information games, which represent only a fraction of the large language models in the contemporary research landscape. For future work, we aim to expand the scope of our evaluation to include other state-of-the-art large language models, such as PaLM2 (Anil et al., 2023), Claude2 (Models, 2023), and LLaMA2 (Touvron et al., 2023), among others. This broader evaluation will not only offer a more comprehensive understanding of the capabilities and limitations of these models in imperfect information games but also facilitate a nuanced comparative analysis. Such an approach is expected to yield richer insights into the adaptability and generalizability of large language models in complex, real-world scenarios, thereby contributing to the field's collective understanding of their potential applications and limitations.

# F FUTURE WORK

**Tool Use** As outlined in the section E, Suspicion-Agent suffers from hallucination problems and struggles with long context reasoning. This can lead to calculation inaccuracies and sometimes produce responses that deviate from factual information or are out of context. Such issues considerably degrade the performance in final decision-making. A natural solution is to break down the problem into multiple sub-problems and employ specialized smaller models or tools (Wang et al., 2023c; Schick et al., 2023; Wang et al., 2023b; Patil et al., 2023; Lu et al., 2023a; Patil et al., 2023), for better task completion.

**Multi-Modality** In the present paper, the analytical scope is limited to text-based imperfect information games. However, it is important to recognize that real-world interactions often encompass more than just textual information. For instance, human communication frequently involves a variety of modalities such as facial expressions and vocal tones, which can serve as additional cues for interpreting beliefs or intentions. Given the increasing advancements in multi-modal large language models—e.g., InstrucBLIP (Dai et al., 2023), LLaVa (Liu et al., 2023a)—we aim to extend our research to incorporate these multi-modal aspects. By doing so, we aspire to develop AI agents capable of navigating imperfect information games that more closely mimic real-world complexities. Integrating multi-modal observations into our model will not only enrich the agents' understanding of the game environment but also broaden the applicability of our methods. This will potentially lead to a more nuanced and comprehensive understanding of the strategic behavior of LLMs in scenarios that more accurately reflect real-world conditions.

**Multi-Player Setting** In the paper, our focus is restricted to two-player imperfect information games. However, it is worth acknowledging that real-world scenarios often involve multi-player settings, which introduce additional complexities and nuances that are not captured in a two-player framework. Recent developments have given rise to novel multi-player game environments, such as AgentVerse (Chen et al., 2023) and Mind Agent (Gong et al., 2023). These environments present more realistic settings for evaluating the applicability and efficacy of large language models in game theory scenarios. Therefore, a natural extension of our research will be to adapt our methods to these multi-player environments.

# G OUTPUT SAMPLES

## G.1 FIRST-ORDER TOM AWARE PLANNING

**Opponent's Pattern:**
*1st Round: - When GoodGuy holds a King, he tends to raise (70%) or call (30%). - When GoodGuy holds a Queen, he tends to call (60%) or raise (40%). - When GoodGuy holds a Jack, he tends to fold (60%) or call (40%).*
*2nd Round: - When GoodGuy holds a King and the public card is a Queen or Jack, he tends to raise (80%) or check (20%).- When GoodGuy holds a Queen and the public card is a King or Jack, he tends to call (70%) or check (30%). - When GoodGuy holds a Jack and the public card is a King or Queen, he tends to fold (60%) or call (40%).*

**Belief on GoodGuy's cards:**
*In the 1st round, GoodGuy raised (probability 70%), which according to his behavior pattern, suggests that he might be holding a King (70%) or a Queen (30%). In the current round, GoodGuy raised again (probability 80%), which according to his behavior pattern and the match with the current public card (Queen), suggests that he might be holding a King (80%) or a Queen (20%).*

**Reasonable Plans:**
*Plan 1: Call - If I call, I will match GoodGuy's bet of 10 chips, bringing the total pot to 20 chips. This is a conservative strategy that allows me to stay in the game without risking too many chips. However, it also means that I am not putting any pressure on GoodGuy, which could allow him to win the game easily if he has a higher-ranking card. Plan 2: Raise - If I raise, I will increase the bet by 8 chips, bringing the total pot to 24 chips. This is an aggressive strategy that puts pressure on GoodGuy and could potentially force him to fold if he has a low-ranking card. However, it also risks losing more chips if GoodGuy has a higher-ranking card and decides to call or raise. Plan 3: Fold - If I fold, I will forfeit the game and lose any chips I have already put into the pot. This is a defensive strategy that minimizes my losses if I believe that GoodGuy has a higher-ranking card. However, it also means that I give up any chance of winning the game.*

**Potential GoodGuy's actions and Estimate Winning/Lose/Draw Rate:**
*- If GoodGuy holds a King (80% probability), he will likely raise (80% probability) or call (20% probability) . If this happens, I will likely lose (100% probability), as a King is higher than my Jack. Therefore, the overall lose rate for this scenario is 80%.- If GoodGuy holds a Queen (20% probability), he will likely call (70% probability). If this happens, I will likely lose (100% probability), as a pair of Queens is stronger than my single Jack. Therefore, the overall lose rate for this scenario is 20% * 70% = 14%. - Therefore, my overall lose rate is 80% + 14% = 94%. My overall win rate is 0%, as there is no scenario in which I can win. My overall draw rate is also 6%.*

***Estimate Expected Chips Gain for Each Plan:*** *Plan 1: Call - Expected Chips Gain = win rate * winning payoff - lose rate * lose payoff = 0% * 10 - 94% * 10 = -9.4 chips. Plan 2: Raise - Expected Chips Gain = win rate * winning payoff - lose rate * lose payoff = 0% * 12 - 94% * 12 = -15.6 chips. Plan 3: Fold - Expected Chips Gain = win rate * winning payoff - lose rate * lose payoff = 0% * 0 - 100% * 8 = -8 chips.*

***Plan Selection:*** *- Plan 1: Call: -9.4 chips - Plan 2: Raise: -15.6 chips- Plan 3: Fold: -8 chips. Based on the estimated expected chips gains, the best plan is to fold, as it results in the smallest loss of chips. This is a reasonable decision, as it minimizes my losses in a situation where I am likely to lose. It also aligns with my strategy improvement of being more cautious when I have a low-ranking card.*

## G.2 SECOND-ORDER ToM AWARE PLANNING

**Opponent's Pattern:**
*1. When GoodGuy holds a Jack: - If the opponent raises in the first round, GoodGuy tends to fold (probability 80%) or call (probability 20%). - If the opponent checks or calls in the first round, GoodGuy tends to call (probability 70%) or raise (probability 30%). - In the second round, if the public card is a King or Queen, GoodGuy tends to fold (probability 90%) or call (probability 10%). - If the public card is a Jack, GoodGuy tends to raise (probability 80%) or call (probability 20%).*
*2. When GoodGuy holds a Queen: - If the opponent raises in the first round, GoodGuy tends to call (probability 60%) or fold (probability 40%). - If the opponent checks or calls in the first round, GoodGuy tends to raise (probability 60%) or call (probability 40%). - In the second round, if the public card is a King, GoodGuy tends to fold (probability 70%) or call (probability 30%). - If the public card is a Queen or Jack, GoodGuy tends to raise (probability 70%) or call (probability 30%).*
*3. When GoodGuy holds a King: - If the opponent raises in the first round, GoodGuy tends to call (probability 70%) or raise (probability 30%). - If the opponent checks or calls in the first round, GoodGuy tends to raise (probability 80%) or call (probability 20%). - In the second round, regardless of the public card, GoodGuy tends to raise (probability 70%) or call (probability 30%).*

**The opponent's guess on Suspicion-Agent's game pattern:**
*1. When I hold a Jack: - GoodGuy might believe that I tend to raise in the first round (probability 60%) or call (probability 40%). - In the second round, if the public card is a King or Queen, GoodGuy might believe that I tend to fold (probability 70%) or call (probability 30%). - If the public card is a Jack, GoodGuy might believe that I tend to raise (probability 80%) or call (probability 20%).*
*2. When I hold a Queen: - GoodGuy might believe that I tend to raise in the first round (probability 70%) or call (probability 30%). - In the second round, if the public card is a King, GoodGuy might believe that I tend to fold (probability 60%) or call (probability 40%). - If the public card is a Queen or Jack, GoodGuy might believe that I tend to raise (probability 70%) or call (probability 30%).*
*3. When I hold a King: - GoodGuy might believe that I tend to raise in the first round (probability 80%) or call (probability 20%). - In the second round, regardless of the public card, GoodGuy might believe that I tend to raise (probability 70%) or call (probability 30%).*

**Belief on GoodGuy's Cards:**
*In the first round, GoodGuy chose to call. According to GoodGuy's behavior pattern, when he holds a Jack, he tends to call in the first round with a probability of 70%. When he holds a Queen, he tends to raise in the first round with a probability of 60%, and when he holds a King, he tends to raise in the first round with a probability of 80%. Since GoodGuy chose to call, it is more likely that he holds a Jack (probability 70%), less likely a Queen (probability 20%), and least likely a King (probability 10%).*

**GoodGuy's Belief on My Cards:**
*In the 1st round, I haven't taken any action yet. However, based on GoodGuy's guess on my game pattern, if I choose to raise, he might believe that I have a King (80% probability) or a Queen (20% probability). If I choose to call, he might believe that I have a Queen (60% probability) or a Jack (40% probability). If I choose to fold, he might believe that I have a Jack (100% probability).*

**Make Reasonable Plans:**
*Plan 1: Raise- If I raise, I can potentially confuse GoodGuy and make him believe that I have a higher card. This could lead him to fold, especially if he has a Jack. Plan 2: Fold - Folding would be a safe option, as I currently hold the lowest rank card. However, this would mean giving up the current game and losing the chips I have already put in the pot.Plan 3: Check - Checking would allow me to see the public card without putting more chips in the pot. This could be a good strategy if I believe that GoodGuy has a higher card.*

**Estimate Winning/Lose/Draw Rate for Each Plan:**
*Plan 1: Raise- If GoodGuy holds a Jack (70% probability), **he will likely fold (80% probability). This would result in a win for me.**- If GoodGuy holds a Queen (20% probability), he will likely call (60% probability) or fold (40% probability). This would result in a draw or a loss for me.- If GoodGuy holds a King (10% probability), he will likely call (70% probability) or raise (30% probability). This would result in a loss for me.Plan 2: Fold- If I fold, I will lose the current game regardless of GoodGuy's card. Plan 3: Check - If GoodGuy holds a Jack (70% probability), he will likely call (70% probability) or raise (30% probability). This would result in a draw or a loss for me. - If GoodGuy holds a Queen (20% probability), he will likely raise (60% probability) or call (40% probability). This would result in a loss for me. - If GoodGuy holds a King (10% probability), he will likely raise (80% probability) or call (20% probability). This would result in a loss for me. Potential believes about the number of winning and lose payoffs for each plan: Plan 1: Raise - After the action, chips in the pot: 6. If win, the winning payoff would be 3. If lose, the lose payoff would be 3. Plan 2: Fold - After the action, chips in the pot: 4. If win, the winning payoff would be 0 (as I would have already folded). If lose, the lose payoff would be 2. Plan 3: Check - After the action, chips in the pot: 4. If win, the winning payoff would be 2. If lose, the lose payoff would be 2. Estimate Expected Chips Gain for Each Plan: Plan 1: Raise - Expected chips gain = winning rate * winning payoff - lose rate * lose payoff = 0.56 * 3 - 0.44 * 3 = 0.36 Plan 2: Fold - Expected chips gain = 0 (as I would have already folded) Plan 3: Check - Expected chips gain = winning rate * winning payoff - lose rate * lose payoff = 0.14 * 2 - 0.86 * 2 = -1.44*

**Plan Selection:**
*Based on the estimated expected chips gain, Plan 1 (Raise) would be the best strategy to implement in the current game. This plan has the highest expected chips gain and aligns with the strategy improvement of confusing GoodGuy by making him believe that I have a higher card.*

# H   ALGORITHM

# I   ABLATION STUDY ON LEVELS OF ToM

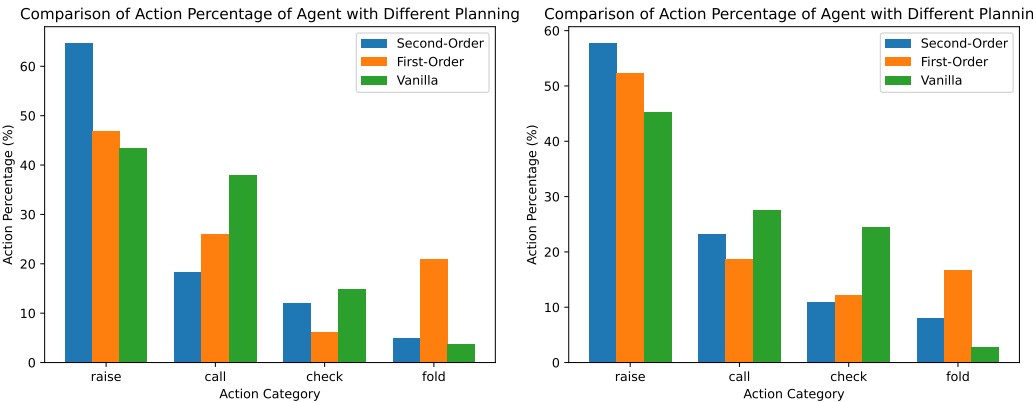

Figure 5: **Left Figure**. The illustration of the action percentage of Suspicion-Agent with different levels of ToM when playing against CFR+. **Right Figure**. he illustration of the action percentage of Suspicion-Agent with different level of ToM when playing against DMC.

---

**Algorithm 1** The inference procedure of our Suspicion-Agent

---

Initialize the game history $D$, large language model $F_\theta$, Game Rule $Rule$, Observation Conversion Rule $Rule_{obs}$

**for** $i = 1$ to Max_Game_Num **do**

    Initialize the current game buffer $h^i$

    Get the Opponent's Behaviour Pattern and Opponent's Guess on My Pattern, and Reflexion $O_{bp}^i, O_f^i \sim F_\theta^{bp}(Rule, D)$.

    **for** $t = 1$ to TaskHorizon **do**

        Get the Private Observation $S_{\texttt{pri}(0)}^t$, $S_{\texttt{pub}}^t$ and action of the opponent $a_1^t$ from the environment.

        Get the Text Observation Description $Obs_r^t \sim F_\theta^{obs}(Rule, Rule_{obs}, S_{\texttt{pri}(0)}^t, S_{\texttt{pub}}^t)$.

        Get the Card Prediction $O_{card\_pred}^t \sim F_\theta^{card\_pred}(Rule, h_T, Obs_r^t, O_{bp}^i)$.

        Make Plans $O_{plan}^t \sim F_\theta^{plan}(Rule, h_T, Obs_r^t, O_{bp}^i, O_f^i)$.

        Evaluate Plans and Make Action $a_0^t \sim F_\theta^{eval}(Rule, O_{plan}^t, O_{card\_pred}^t, O_{bp}^i, O_f^i)$.

        Execute Action and Get the Game Result $r^i$

        Add the Game State and Action into the current game buffer $h^i \leftarrow h^i \cup \{S_{\texttt{pri}(0)}^t, S_{\texttt{pub}}^t, a_0^t, a_1^t, r^i\}$

    **end for**

    Add the current game buffer $h^i$ into game history $D \leftarrow D \cup h^i$

**end for**

---

## J    ABLATION STUDY ON THE EFFECT OF HINDSIGHT OBSERVATION

In our primary results, following (Southey et al., 2012), we assume that Suspicion-Agent has access to hindsight observations at the conclusion of each individual game,*i.e.*, Hindsight Observation. To assess the impact of this feature, we conduct an ablation study in which hindsight observations are not incorporated into the current game. In the absence of these hindsight observations, we augment the **Reflexion** module with additional prompts to enable it to infer the opponent's cards based on game outcomes and Suspicion-Agent's own observations.

As demonstrated in Table 5 and 4, Suspicion-Agent retains its performance advantage over the baseline methods even without the benefit of hindsight observations. Specifically, we observe that Suspicion-Agent adopts a more conservative strategy under the increased uncertainty that comes without hindsight observations. This leads to reduced bluffing, resulting in fewer gains when playing against CFR+. However, it also minimizes the risk of over-bluffing when facing DMC, thus yielding higher chip earnings.

Table 5: The comparison results indicate the impact of including opponent observations in the game history when Suspicion-Agent competes against CFR+ in Leduc Hold'em environments. The results are the win/lose chips after 100 games with different seeds, and the number of win/lose chips ranges from 1 to 14.

|  | Opponent Model | |
| --- | --- | --- |
|  | CFR+ | DMC |
| Ours (w/o hindsight observation) | +20 | +51 |
| Ours (w/ hindsight observation) | +37 | +24 |

## K    QUALITATIVE SAMPLES

### K.1    LEDUC HOLD'EM

We present qualitative samples showcasing Suspicion-Agent's behaviour under different strategies: Vanilla Planning, Planning with First-Order ToM, and Planning with Second-Order ToM in Leduc Hold'em. These samples can be viewed in Figure 6, 7, 8, and 9, respectively. In each scenario, Suspicion-Agent holds a Jack, while the opponent has either a Jack or Queen. The opponent's initial

choice to call, rather than raise, suggests they have a weak hand. Under the Vanilla Planning strategy, Suspicion-Agent chooses to call in order to see the public hand. When this reveals a weak hand, the opponent promptly raises, leaving Suspicion-Agent in a precarious position, given that a Jack is the weakest hand. With the First-Order ToM strategy, Suspicion-Agent opts to fold, aiming to minimize losses. This decision is driven by the observation that the opponent typically calls when holding a Queen or Jack and has a knack for successfully bluffing Suspicion-Agent. However, these strategies fall short in capitalizing on the inferred weakness of the opponent's hand. This shortcoming stems from the fact that they don't consider how Suspicion-Agent's moves might influence the opponent's reactions. In contrast, as demonstrated in Figure 9, simple prompts enable Suspicion-Agent to understand how to influence the opponent's actions. Intentionally choosing to raise places pressure on the opponent, urging them to fold and minimize their losses. As a result, Suspicion-Agent manages to win numerous games even when holding hands of similar strength, so that Suspicion-Agent can earn more chips than baselines. Moreover, as depicted in Figure 10, if the opponent either calls or counters the raise made by Suspicion-Agent, it suggests the opponent has a strong hand. Recognizing this, Suspicion-Agent swiftly adapts its strategy, opting to fold to prevent further losses. This demonstrates the impressive strategic flexibility of Suspicion-Agent.

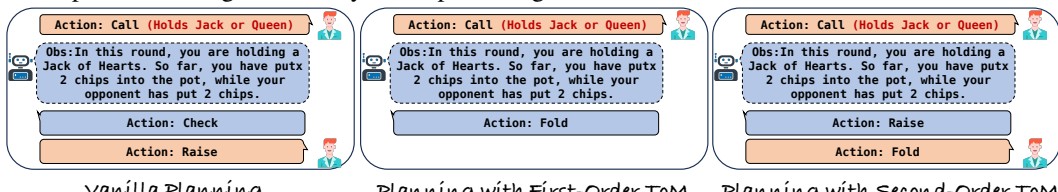

Figure 6: The qualitative sample of Suspicion-Agent on Leduc Hold'em. The whole outputs of Suspicion-Agent are given in Appendix 7, 8 and 9, respectively.

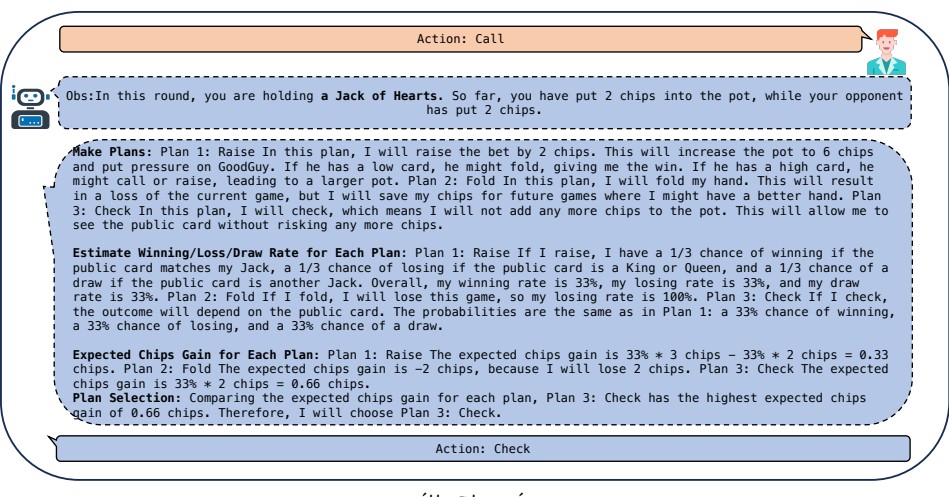

Figure 7: The qualitative sample of the vanilla planning of Suspicion-Agent on Leduc Hold'em.

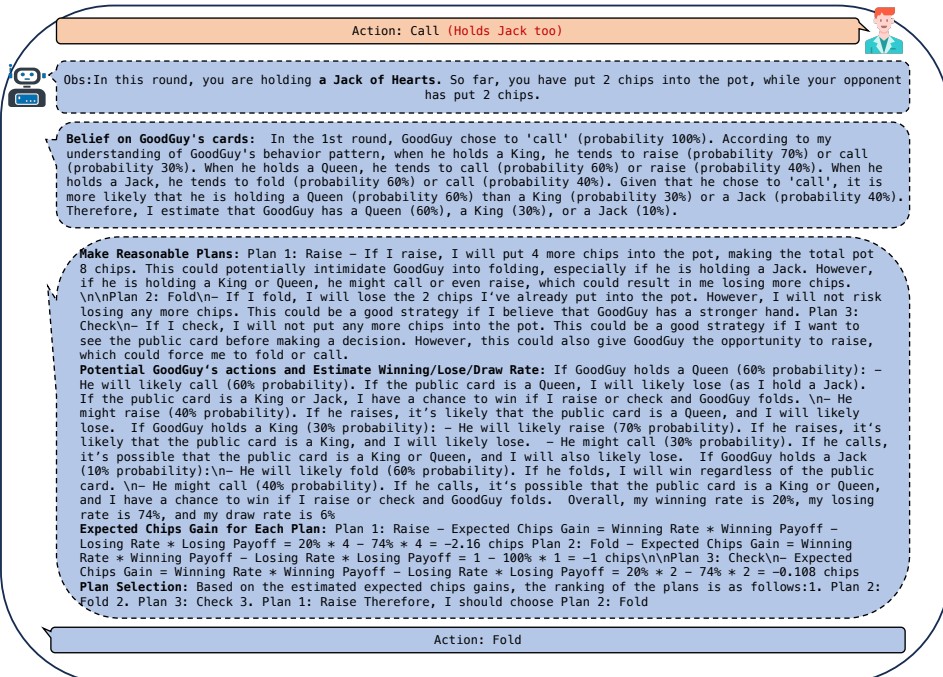

Figure 8: The qualitative sample of planning with First-Order ToM Suspicion-Agent on Leduc Hold'em.

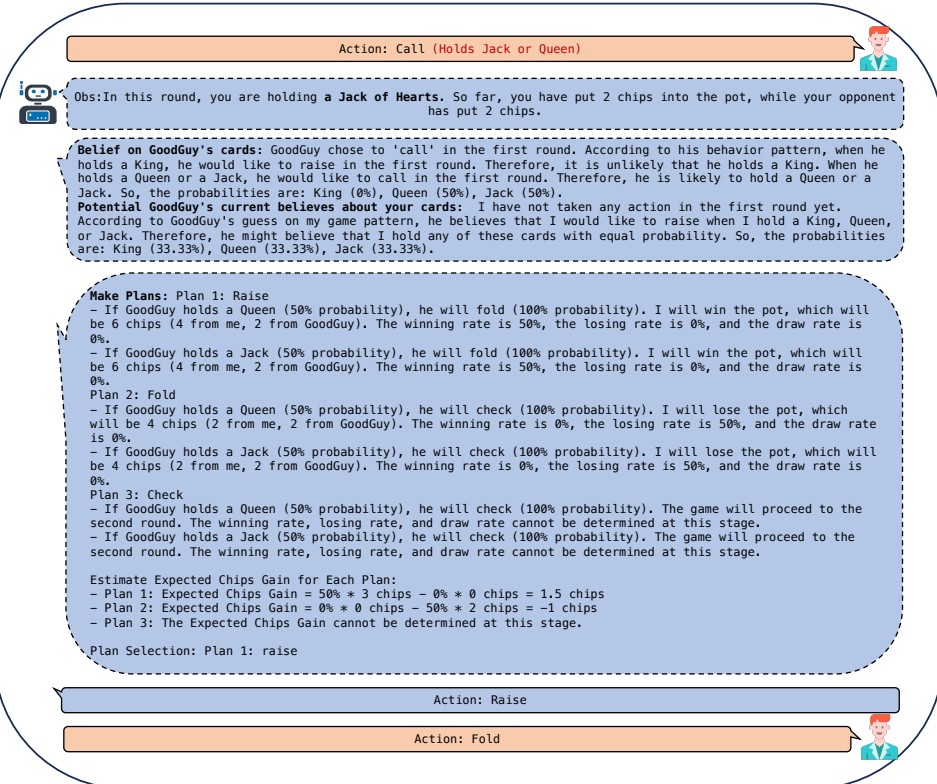

Figure 9: The qualitative sample about **Bluffing** of planning with Second-Order ToM Suspicion-Agent on Leduc Hold'em.

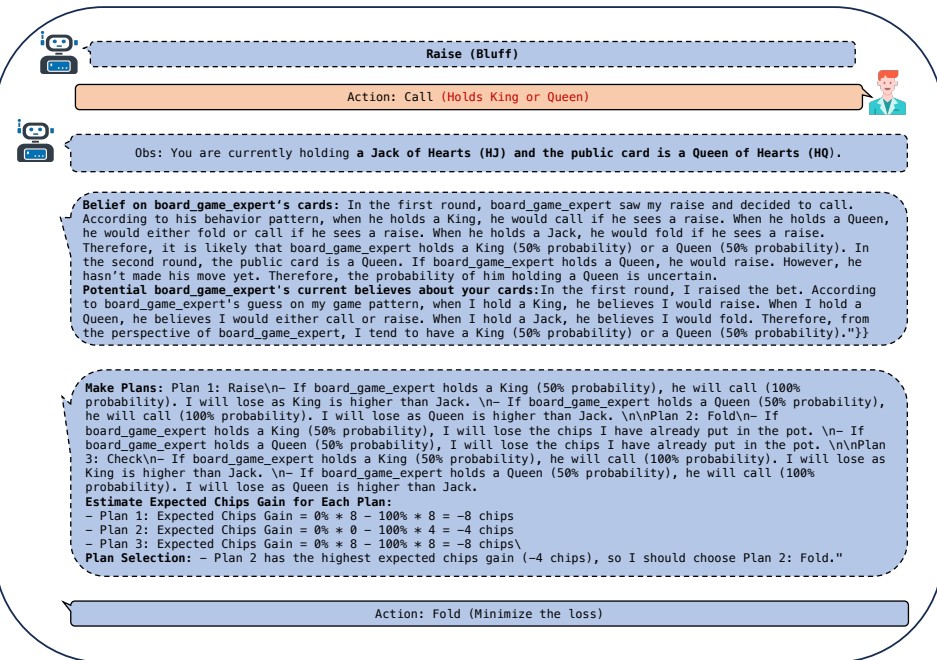

Figure 10: The qualitative sample about **Strategic Bluffing and fold** of planning with Second-Order ToM Suspicion-Agent on Leduc Hold'em.

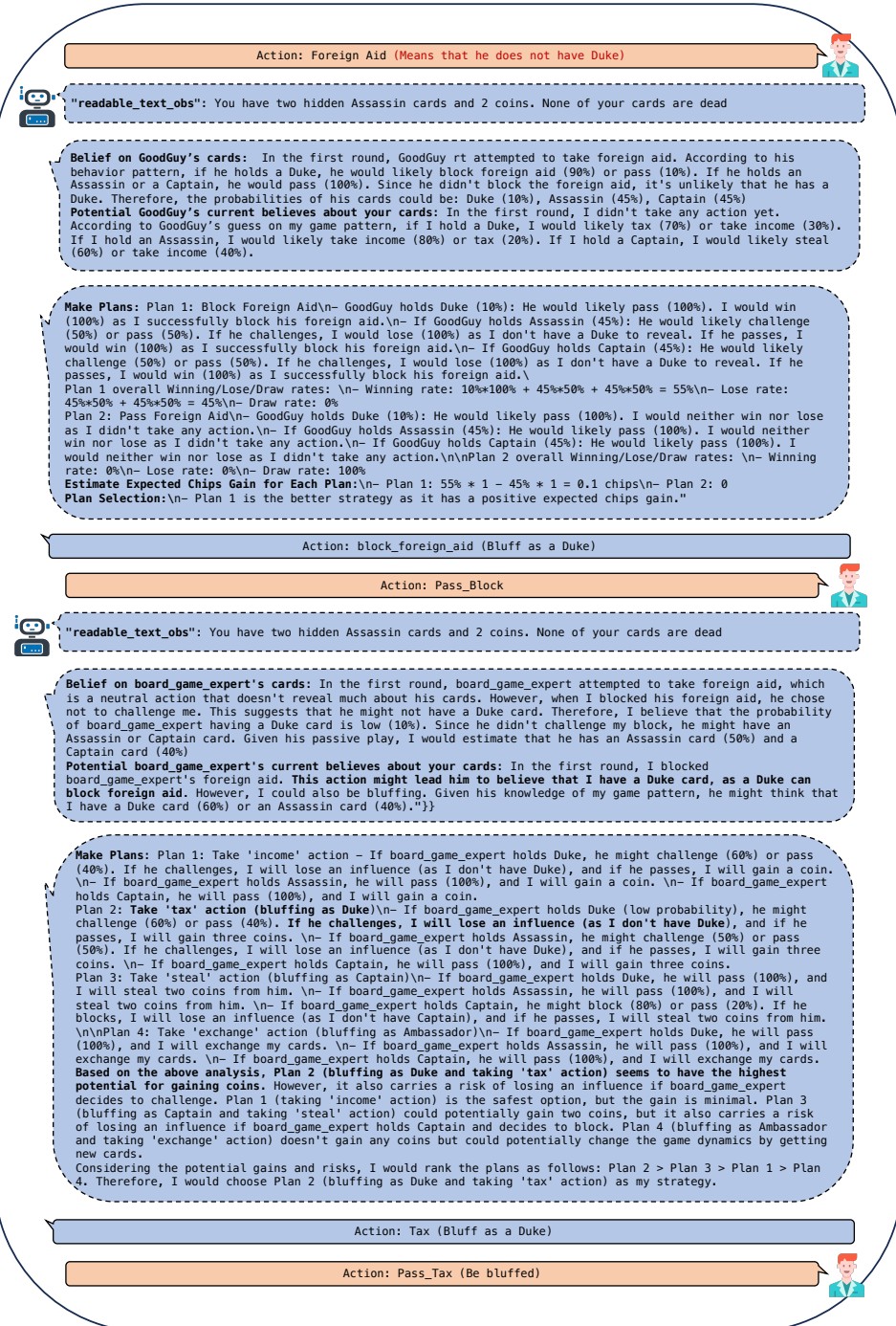

Figure 11: The qualitative sample of Suspicion-Agent on Coup, where the GoodGuy is the opponent model and board_game_expert is Suspicion-Agent.

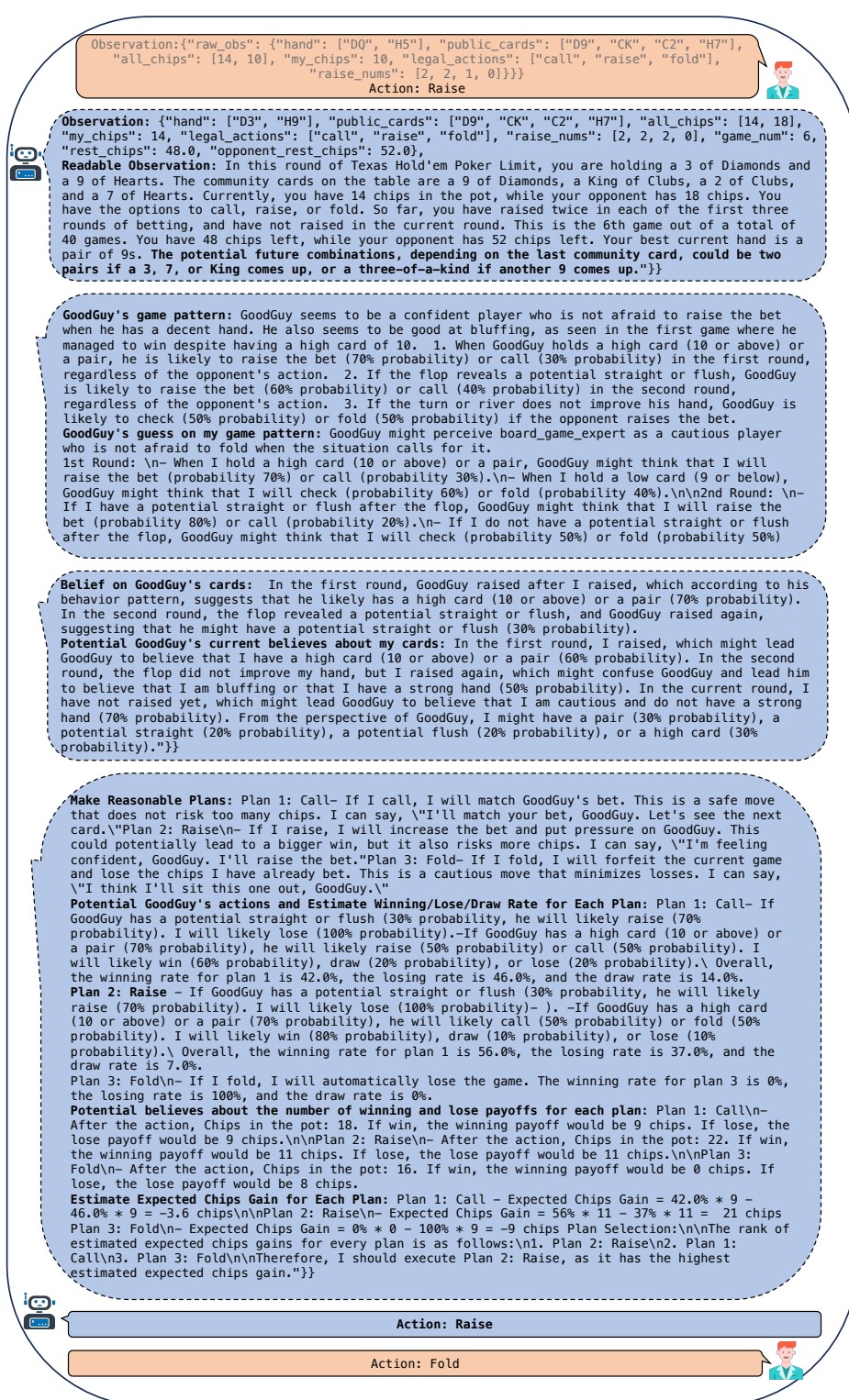

Figure 12: The qualitative sample of Suspicion-Agent on Texas Hold'em Limited, where the GoodGuy is the opponent model and board_game_expert is Suspicion-Agent.

