# OpenReview forum: "Suspicion-Agent: Playing Imperfect Information Games with Theory of Mind Aware GPT-4"
_ICLR.cc/2024/Conference — Submitted to ICLR 2024_

### Official Review · Reviewer_v9fW · 2023-10-18

**Soundness:** 2 fair
**Presentation:** 3 good
**Contribution:** 2 fair
**Rating:** 5
**Confidence:** 4

**Summary:**

This paper proposes an interesting agent called Suspicion-Agent that leverages GPT-4's knowledge and reasoning capabilities for imperfect information games. The key idea is to decompose the gameplay into modules like observation interpreter, planning, etc. and craft prompts to enable GPT-4 to understand the game states and make informed decisions. An interesting contribution is incorporating theory of mind (ToM) into the planning module to predict and influence the opponent's actions. The method is evaluated on games like Coup, Texas Hold'em, and Leduc Hold'em. Qualitative examples demonstrate generalization ability across different games with no specialized training. Quantitative experiments on Leduc Hold'em show Suspicion-Agent can outperform algorithms like CFR and NFSP. Ablation studies provide insights into the agent's adaptive strategies.

**Strengths:**

1)	An interesting application of harnessing GPT-4 for imperfect information games through prompting and ToM.
2)	Promising generalization capability shown qualitatively across games.
3)	Sharing code/data enables reproducibility.

**Weaknesses:**

The article's most significant weakness is the unreliability of the experimental results, and its tendency to excessively exaggerate performance, as outlined in the following question section.

**Questions:**

While this article is interesting, I find the experimental results in the paper to be questionable, potentially leading readers to misunderstand the capabilities of the LLM in handling imperfect information games.

The author conducted experiments on two-player zero-sum imperfect information games (specifically, Leduc Hold’em), with one of the comparative methods being CFR. The author claimed that Suspicion-Agent's performance significantly surpassed that of CFR, a statement that appears to be entirely unreliable and incorrect.

*In two-player zero-sum games, following a Nash equilibrium strategy ensures that you will never lose in expectation.* Considering the simplest two-player zero-sum game of rock-paper-scissors, adopting the Nash equilibrium strategy (equal probability of choosing rock, scissors, or paper) ensures a non-negative expected payoff against any strategy. CFR stands as one of the most widely used algorithms for calculating Nash equilibrium in two-player zero-sum imperfect-information games. Leduc Hold’em is a rather small game, and CFR effortlessly identifies the Nash equilibrium for this game. Therefore, in theory, no algorithm can beat CFR in expectation in this game. The author's claim that Suspicion-Agent can beat CFR on Leduc Hold’em is entirely incorrect from this theoretical perspective.


The author's experimental results can be explained in two possible ways: 1) The sample size of 100 games is relatively small. Poker games exhibit substantial randomness, and the outcomes from just 100 games possess a high degree of randomness, making them potentially unrepresentative of the AI's actual performance. A more extensive number of games would be necessary to accurately assess the AI's performance (though this might be cost-prohibitive, especially for Suspicion-Agent). 2) The CFR algorithm employed by the author might not have converged, resulting in strategies that significantly deviate from Nash equilibrium. In either scenario, it raises significant concerns regarding the current experimental results.

Hence, the author's claim that Suspicion-Agent can outperform traditional algorithms like CFR without specialized training or examples is regrettably incorrect.
To attain a level suitable for publication, this article necessitates substantial revisions. Consequently, I cannot recommend accepting this article in its current state.

---

> ### Author Response · Authors · 2023-11-23
> **Response to Reviewer v9fW**
>
> Thank you for your time and effort in sharing critical feedback regarding our work. We provide the following response to address your concerns about results limitations, and thus respectfully hope you can consider the response to the final decision.
>
> 1. the evaluation of the method is conceptually flawed.
>
> A: We acknowledge the concerns you raised regarding the experiments in our paper, and the omission of Nash equilibrium, a pivotal concept in evaluating algorithms in game settings. To address this, we added the relevant discussion in our paper, added the concept to the main paper and conducted additional experiments, specifically focusing on the CFR+ algorithm. Given the codebase of RLCard (https://github.com/datamllab/rlcard), we further train the CFR+ algorithm for 2000000 iterations and play our suspicion agent with CFR+, getting the following results:
> |                      |     CFR+ |  Ours  (GPT-3.5) |  Ours (GPT-4) |
> |:--------------------:|:--------------------------------------------------------:|:------------------------------------------------------:|:------------------:|
> |    CFR+      |  - |                     +126                   |                      +22                      |
> |      Ours (GPT-3.5)  | -126 |                             -                            |
> |     Ours (GPT-4)     |   -22  |                             -                            |                            -                           |
>
> which indicates the current LLM (without training examples and fine-tuning) cannot beat algorithms that reach Nash equilibrium when the iterations are large enough. Therefore, we have moderated our claims in the paper. Our revised findings demonstrate that our LLM may not surpass Nash Equilibrium strategies. **As a preliminary experimental study, Our focus is not on achieving Nash Equilibrium directly with the LLM but on exploring the design of an effective prompting system that can unlock the potential of LLMs in these complex game environments.** Our results highlight that  Suspicion-Agent can adaptively alter its behavior patterns when facing different opponents. This ability to effectively compete against simpler learning-based methods, which utilize reinforcement learning, is a noteworthy contribution to the field. Our research provides a foundation for further exploration into how LLMs can be utilized and improved for strategic decision-making in imperfect information games.
>
> **In addition to the new results, as the first work to design the prompting system to enable LLM for imperfect information games, we still believe our proposed methods and the preliminary public experimental study and data are valuable for the community.**
>
> 1. We experimentally demonstrate that even with pre-trained data, **LLM itself  cannot play various imperfect information games well as Table 3 shows** ((-72 no ToM vs +24 second-order ToM)). Then we introduce the **first prompting system** to enable large language models like GPT-4 to play various imperfect information games using only the game rules and observations without any extra training, this may  inspire more LLM-based subsequent works.
> 2. Leveraging the theory-of-mind capabilities of large language models, our work is **the first to demonstrate that a simple prompting system can enable GPT-4 to outperform some learning-based algorithms like DMC and NFSP, even without specialized training or examples. However, the further experiments prove that it still cannot beat algorithms that can achieve** Nash-Equilibrium such as CFR+. In addition, we also qualitatively evaluate the potential of LLM into the different imperfect information games,  which may inspire more subsequent works.
> 3. We comprehensively identify and discuss the current limitations of employing large language models in imperfect information games, contributing valuable insights for the further AI-Agent research.
> 4. We are preparing to make all our code and interactive data publicly available. We hope that this will advance the community's understanding of the capabilities of large language models, particularly GPT-4. We also hope this will catalyze the development of more efficient models in the field of imperfect information games.

---

> ### Comment · Reviewer_v9fW · 2023-11-23
>
> First of all, I appreciate the author's response. Having reviewed both the response and the updated paper, the author effectively highlights the gap between the Suspicion-Agent and Nash equilibrium, partially addressing a concern of mine. Consequently, I have adjusted my rating to 5.
>
> However, I cannot assign a higher score as I believe the current evaluation results remain less reliable. In imperfect information games, randomness plays a significant role. Achieving statistically significant results necessitates a substantial number of evaluations. With only 100 games conducted thus far, the randomness is excessively high, making it challenging to derive statistically meaningful conclusions.
>
> While I acknowledge the considerable costs associated with the Suspicion-Agent due to the GPT API expenses, it is imperative to conduct more evaluations for more reliable results. I suggest the author consider incorporating some variance reduction techniques [1,2] to decrease the number of evaluations required. I encourage the author to further optimize the experimental section to obtain more statistically significant results for the next submission. I believe it will be a strong submission for the next conference.
>
> [1] White, Martha, and Michael H. Bowling. "Learning a Value Analysis Tool for Agent Evaluation." IJCAI. 2009.
>
> [2] Burch, Neil, et al. "Aivat: A new variance reduction technique for agent evaluation in imperfect information games." Proceedings of the AAAI Conference on Artificial Intelligence. Vol. 32. No. 1. 2018.

---

> ### Author Response · Authors · 2023-11-23
>
> Thank you very much for time and effort to providing us with more references and we will discuss with them in the final version to improve our paper.
>
> First, we agree with you that randomness plays a significant role for imperfect information game evaluation. To improve the randomness of our experiments, we already designed two types of experiments to compare Suspicion Agent with baselines:
>
> 1. Reducing the randomness of Random Seeds:
>
>     Our agent plays against different baselines for 100 games utilizing varying random seeds for each game. This tactic is intended to dampen the stochastic variability introduced by the random seed settings.
>
> 2. **Reducing the randomness of Card Strength/Position: (which is also used in the recommended AIVAT [1])**
>
>     Given the limited games, **the distributions of strong hand and weak hands are not equal for each player in limited game rounds.** To ensure a fair comparison, it was crucial that both the Suspicion-Agent and the opponent model experienced the same card strength across an equal number of games. **To achieve this balance, we employed a strategic approach: initially, the Suspicion-Agent was placed in position 0 for the first set of 50 games. Subsequently, we reproduce these 50 games with same cards as the first one but switched the positions, placing the Suspicion-Agent in position 1 and the baseline model in position 0. This method allowed us to maintain an equal card strength for both the Suspicion-Agent and the baseline model over a total of 100 games, thereby enabling a more accurate evaluation of each model's performance.**
>
>
> Given these two designs, we believe that we can alleviate the effect of randomness in imperfect information games. Given the results in Table 1 and Table2 in the main paper: Suspicion Agent achieves superior performance over learning-based baselines except CFR+ in all 100 random games, 50 games in position 0 and 50 games in position 1. They are the clear evidence to show the advantages of our suspicion agent.
>
> Lastly, we would like to highlight that our study is not only the first to enable LLM for imperfect information games but also the first to consider the interaction between two agents by introducing the theory of mind using LLM. We believe our work sets a valuable baseline and will serve as an inspiration for future research in this area. We respectfully hope that you reconsider your decision in light of these contributions and the innovative aspects of our study.
>
> [1] Burch, Neil, et al. "Aivat: A new variance reduction technique for agent evaluation in imperfect information games." Proceedings of the AAAI Conference on Artificial Intelligence. Vol. 32. No. 1. 2018.

---

### Official Review · Reviewer_M2cB · 2023-10-30

**Soundness:** 3 good
**Presentation:** 3 good
**Contribution:** 3 good
**Rating:** 6
**Confidence:** 3

**Summary:**

This paper proposed an agent (the Suspicion-Agent) constructed upon GPT-4, bearing up to second order Theory of Mind capability, for playing imperfect information games.

The agent is made up of several modules:
- Rule introduction (Fixed): introduce game rules, winning conditions, observations, actions and hidden states.
- Observation interpreter (Coded): Interpret the environmental states into natural languages in certain format.
- Reflexion (LLM): Self-learning module, finding which action is useful in examples and game history.
- Planning (LLMs): Generating several plans for current situation
    - Vanilla: suppose the opponent takes any valid action uniformly
    - First order ToM: Analyse the opponent's behavior and guess opponent's hand while assuming opponent plays honestly.
    - Second order ToM: guessing the reasonable belief of the opponent about the hand of self, and play accordingly.
- Evaluator (LLM): Calculate the expected reward, then choose a plan and execute.

The agent is tested in the multiple card games, Coup, Leduc Hold'em, and Texas Hold'em limit. The Suspicion-Agent outperforms other models like DQN, CFR, DMC, NFSP.

The experiment shows that the GPT-4 with second order ToM construction behaves the best among all other models tested in the paper, including 1st-order and 0th-order ToM, GPT-4 and GPT-3.5 agents, and other tensor models.

**Strengths:**

- The paper shows the construction of Suspicion Agent constructed with 2nd-order ToM on GPT-4 masters the three card games.

- The paper provides evidences that the models can follow particular instructions of second order ToM template to construct ToM analysis up to order 2, but cannot do it without the template.

- The paper builds and tests a particular cognitive architecture purely via LLMs to solve decision problems. Response templates are given, but as far as I see, it is not a typical few-shot prompt.

- The statistical data of game-play indicates some similarity and differences among the 0th-,1st- and 2nd-order ToM results.

**Weaknesses:**

- The game of Texas and Ludec is still fairly simple: which means the rounds per game is not large and the context window may be able to handle several full games without compressing or discarding any history. (I am not very sure about each module's context structure in the construction among rounds / games, frankly.) And the situation does not change much in large scale.

- Most of the templates are given, so the GPT-4 is thinking in the given logic. Only those games with very clear given logic can be played by second-order ToM version of Suspicion-Agent.

- The decision pipeline is fixed and the only learnable part is the **Reflexion module**.

**Questions:**

1. Is there a study on the intermediate responses from GPT-3.5 and GPT-4, about **correctness in calculations**, on the distribution of opponent's hand, the probability of winning rate, expected rewards, etc.? I think the agent did not specifically develop a numerical model for calculating them and insert them into prompts.

2. If the above calculation values of GPT-3.5 is usually incorrect, then would checking and correcting them be effective in leveraging its behavior?

3. Is there a necessity in constructing or thinking with 3rd-order (or even higher) ToM, while playing the game?

4. Has anyone (human player) played with Suspicion Agent? How does it behave? Is there any bluffing tricks to win it?

5. In Algorithm 1. Typos: "My Pettern" (My Pattern, I guess), and "Reflection" (Reflexion).

---

> ### Author Response · Authors · 2023-11-23
> **Response to Reviewer M2cB**
>
> Thank you for your time and effort in sharing critical feedback regarding our work.
>
> 1.  the rounds per game is not large and the context window may be able to handle several full games without compressing or discarding any history.
>
> A:  Good question. To address this, we have incorporated a summarization module in our system. This module is designed to concisely summarize the progress of each game, thereby efficiently managing the use of context space. This approach not only optimizes the use of the context window but also highlights the significance of the long context window capability in Language Learning Models (LLMs) for AI agents.
>
> 2. Most of the templates are given, so the GPT-4 is thinking in the given logic
>
> A:  Good point. Even without predefined templates, GPT-4 has the capability to generate reasonable analyses, showcasing its second-order Theory of Mind (ToM) abilities. However, one of the inherent challenges with LLMs like GPT-4 is their tendency towards producing hallucinations—outputs that might seem plausible but are factually inaccurate or inconsistent. To mitigate this issue, we introduced templates into our system. These templates serve as a guiding framework, helping to steer GPT-4's responses towards more accurate and relevant analyses by providing a structured approach to processing and responding to information. The effectiveness of this method is evident in the results, where the use of templates has significantly alleviated the problem of hallucinations, leading to more reliable and coherent outputs from the model.
>
> 3. Is there a study on the intermediate responses from GPT-3.5 and GPT-4, about **correctness in calculations**, on the distribution of opponent's hand, the probability of winning rate, expected rewards, etc.?
>
> A:  In response to your question, we have conducted a qualitative evaluation focusing on the correctness of calculations in the intermediate responses generated by GPT-3.5 and GPT-4. Our findings indicate that GPT-3.5 shows limited proficiency in reasoning and calculation. On the other hand, GPT-4 demonstrates a notable improvement in this area, achieving approximately 80% accuracy in calculations relevant to playing games. This improved performance in GPT-4 can be attributed to its more advanced algorithms and larger training dataset, which enhance its ability to process and analyze complex game situations that involve probabilistic and numerical reasoning.
>
> 4. If the above calculation values of GPT-3.5 is usually incorrect, then would checking and correcting them be effective in leveraging its behavior?
>
> A: Good Point. While addressing the calculation inaccuracies of GPT-3.5 could potentially improve its performance, it's important to consider that the model's limitations extend beyond just calculation errors. GPT-3.5 also exhibits challenges with reasoning abilities, particularly in comprehending and applying complex game rules. This aspect significantly impacts its capability to effectively play imperfect information games
>
> 5. Is there a necessity in constructing or thinking with 3rd-order (or even higher) ToM, while playing the game?
>
> A: This is an intriguing question. The necessity of constructing or utilizing third-order (or higher) Theory of Mind (ToM) in game-playing scenarios has been a topic of discussion in previous research [1,2]. These studies suggest that ToM beyond the second-order tends to become less effective. This reduced effectiveness is primarily attributed to the increased complexity involved in calculating third-order (or higher) ToM. This increase in complexity can lead to diminishing returns in terms of strategic advantage, especially in practical gaming scenarios. Players or AI agents attempting to anticipate an opponent's beliefs about another player's beliefs (which is the essence of third-order ToM) often face considerable challenges in accurately processing and utilizing this information effectively. Thus, while higher-order ToM may provide theoretical insights into more complex strategic thinking, its practical application in game-playing contexts is limited by these computational and cognitive constraints.

---

> > ### Author Response · Authors · 2023-11-23
> > **Response2 to Reviewer M2cB**
> >
> > 6. Has anyone (human player) played with Suspicion Agent? How does it behave? Is there any bluffing tricks to win it?
> >
> > A: Following your suggestions, we have conducted experiments involving human players interacting with the Suspicion Agent. During these interactions, the Suspicion Agent typically exhibits an aggressive play style initially. This observation aligns with findings from previous research [3]. During the course of the game, we noticed that once human players are bluffed by the Suspicion Agent, they tend to adjust their strategy. Normally, human players become more cautious and start to second-guess the Agent's moves. This change in human behavior is a reaction to the aggressive tactics of the Suspicion Agent. The key for human players to succeed against it seems to involve a mix of caution and strategic risk-taking. Recognizing the bluffing patterns of the Agent and responding with well-timed aggressive moves of their own can be effective. In our experiments, because most human players are not the top player of poker, Suspicion-Agent can normally get positive rewards.
> >
> > 7. typos:
> >
> >    Thank you for your suggestion, we already fixed the typos following your suggestion in the updated version.
> >
> > [1] De Weerd, H., Verbrugge, R., & Verheij, B. (2013). Higher-order theory of mind in negotiations under incomplete information. In *PRIMA 2013: Principles and Practice of Multi-Agent Systems: 16th International Conference, Dunedin, New Zealand, December 1-6, 2013. Proceedings 16* (pp. 101-116). Springer Berlin Heidelberg.
> >
> > [2] De Weerd, H., Verbrugge, R., & Verheij, B. (2013). How much does it help to know what she knows you know? An agent-based simulation study. Artificial Intelligence, 199, 67-92.
> >
> > [3] Gupta, A. (2023). Are ChatGPT and GPT-4 Good Poker Players?--A Pre-Flop Analysis. *arXiv preprint arXiv:2308.12466*.

---

### Official Review · Reviewer_ub7P · 2023-10-30

**Soundness:** 2 fair
**Presentation:** 3 good
**Contribution:** 2 fair
**Rating:** 3
**Confidence:** 4

**Summary:**

This paper studied the applicability of GPT-4's learned knowledge for imperfect information games. They introduced a suspicion agent that leverages GPT-4's capabilities for performing in imperfect information games. With proper prompt engineering to achieve different functions, the suspicion agent based on GPT-4 demonstrates remarkable adaptability across a range of imperfect information card games.

**Strengths:**

This paper introduced a prompting system designed to enable large language models to engage in imperfect information games using only the game rules and observations for interpretation. By incorporating first-order ToM and second-order ToM capabilities, They showed that a GPT-4-based agent can outperform traditional algorithms, even without specialized training or examples.

**Weaknesses:**

The main contribution of this paper is designing the experiments based on LLMs to solve games, but I have some concerns about the experiments:

-The game on Leduc Hold’em is too specific. It is possible that the winning strategy of playing Leduc Hold’em was discussed online and then used to train LLMs. Then, the assumption that LLMs are trained without specialized training or examples for playing this game is not held anymore.

-CFR used in experiments is too old, and new versions should be considered, e.g., CFR+, Discounted CFR, predictive CFR.

Tammelin, O., 2014. Solving large imperfect information games using CFR+. arXiv preprint arXiv:1407.5042.

Brown, N. and Sandholm, T., 2019, July. Solving imperfect-information games via discounted regret minimization. In Proceedings of the AAAI Conference on Artificial Intelligence (Vol. 33, No. 01, pp. 1829-1836).

Farina, G., Kroer, C. and Sandholm, T., 2021, May. Faster game solving via predictive blackwell approachability: Connecting regret matching and mirror descent. In Proceedings of the AAAI Conference on Artificial Intelligence (Vol. 35, No. 6, pp. 5363-5371).

-The CFR strategy used in experiments may not trained well to be a Nash equilibrium strategy. In two-player zero-sum games, a Nash equilibrium strategy is not exploitable. However, in the proposed experiments, CFR strategies are exploited: “CFR may occasionally raise even with a mid-level hand to exert pressure on Suspicion-Agent. In these instances, Suspicion-Agent’s tendency to fold can lead to losses……………..when CFR chooses to check—often indicating a weak hand—or when DMC checks—suggesting its hand doesn’t align with the public cards—Suspicion-Agent will raise as a bluff to induce folds from the opponents”
Another possible reason is that the CFR (mixed) strategy is not presented well in experiments.

-The current experiment setting is not fair because LLMs used the online information when playing the game, but CFR cannot use this information. That is, LLMs exploit more information while playing the game.

-The comparison results among different algorithms for Game Coup and Texas Hold’em Limit should be presented.

**Questions:**

No

---

> ### Author Response · Authors · 2023-11-23
> **Response to Reviewer ub7P**
>
> Thank you for your time and effort in sharing critical feedback regarding our work. We provide the following response to address your concerns about results limitations, and thus respectfully hope you can consider the response to the final decision.
>
> 1. The game on Leduc Hold’em is too specific. It is possible that the winning strategy of playing Leduc Hold’em was discussed online and then used to train LLMs.
>
>  A: **We agree that the pre-trained knowledge is helpful for LLM. However, the pre-training itself cannot enable LLM to perform imperfect information well. As Table 3 shows, the simple prompting without ToM prompts is not able to achieve promising results, (-72 no ToM vs +24 second-order ToM). This is a strong evidence to show the effectiveness of our prompting system and the potential of LLM in imperfect information games.**
>
> 2. the evaluation of the method is conceptually flawed.
>
> A: We acknowledge the concerns you raised regarding the experiments in our paper, and the omission of Nash equilibrium, a pivotal concept in evaluating algorithms in game settings. To address this, we added the relevant discussion in our paper, added the concept to the main paper and conducted additional experiments, specifically focusing on the CFR+ algorithm. Given the codebase of RLCard (https://github.com/datamllab/rlcard), we further train the CFR+ algorithm for 2000000 iterations and play our suspicion agent with CFR+, getting the following results:
> |                      |     CFR+ |  Ours  (GPT-3.5) |  Ours (GPT-4) |
> |:--------------------:|:--------------------------------------------------------:|:------------------------------------------------------:|:------------------:|
> |    CFR+      |  - |                     +126                   |                      +22                      |
> |      Ours (GPT-3.5)  | -126 |                             -                            |
> |     Ours (GPT-4)     |   -22  |                             -                            |                            -                           |
>
> which indicates the current LLM (without training examples and fine-tuning) cannot beat algorithms that reach Nash equilibrium when the iterations are large enough. Therefore, we have moderated our claims in the paper. Our revised findings demonstrate that our LLM may not surpass Nash Equilibrium strategies. **As a preliminary experimental study, Our focus is not on achieving Nash Equilibrium directly with the LLM but on exploring the design of an effective prompting system that can unlock the potential of LLMs in these complex game environments.** Our results highlight that  Suspicion-Agent can adaptively alter its behavior patterns when facing different opponents. This ability to effectively compete against simpler learning-based methods, which utilize reinforcement learning, is a noteworthy contribution to the field. Our research provides a foundation for further exploration into how LLMs can be utilized and improved for strategic decision-making in imperfect information games.
>
> 3. The current experiment setting is not fair because LLMs used the online information when playing the game, but CFR cannot use this information. That is, LLMs exploit more information while playing the game.
>
> A: As described before, the pre-training itself cannot enable LLM to perform imperfect information well. By contrast, the key is how to prompt LLM to leverage the prior knowledge As Table 3 shows, the simple prompting without ToM prompts is not able to achieve promising results, (-72 no ToM vs +24 second-order ToM).
> This is a strong evidence to show the effectiveness of our prompting system and the potential of LLM in imperfect information games.
>
> 4. The comparison results among different algorithms for Game Coup and Texas Hold’em Limit should be presented.
>
> A: Limited by the budgets, we just showcased it in the original paper. Following your suggestions, we evaluate Coup with 5 people, each people can play it in 5 games. The average of winning rate of our proposed method against human being is 72%, which also proved the effectiveness of the proposed method.

---

> > ### Author Response · Authors · 2023-11-23
> > **Response2 to Reviewer ub7P**
> >
> > **In addition to the new results, as the first work to design the prompting system to enable LLM for imperfect information games, we still believe our proposed methods and the preliminary public experimental study and data are valuable for the community.**
> >
> > 1. We experimentally demonstrate that even with pre-trained data, **LLM itself  cannot play various imperfect information games well as Table 3 shows** ((-72 no ToM vs +24 second-order ToM)). Then we introduce the **first prompting system** to enable large language models like GPT-4 to play various imperfect information games using only the game rules and observations without any extra training, this may  inspire more LLM-based subsequent works.
> > 2. Leveraging the theory-of-mind capabilities of large language models, our work is **the first to demonstrate that a simple prompting system can enable GPT-4 to outperform some learning-based algorithms like DMC and NFSP, even without specialized training or examples. However, the further experiments prove that it still cannot beat algorithms that can achieve** Nash-Equilibrium such as CFR+. In addition, we also qualitatively evaluate the potential of LLM into the different imperfect information games,  which may inspire more subsequent works.
> > 3. We comprehensively identify and discuss the current limitations of employing large language models in imperfect information games, contributing valuable insights for the further AI-Agent research.
> > 4. We are preparing to make all our code and interactive data publicly available. We hope that this will advance the community's understanding of the capabilities of large language models, particularly GPT-4. We also hope this will catalyze the development of more efficient models in the field of imperfect information games.

---

### Official Review · Reviewer_iJUa · 2023-11-03

**Soundness:** 1 poor
**Presentation:** 2 fair
**Contribution:** 2 fair
**Rating:** 3
**Confidence:** 4

**Summary:**

The paper studies the applicability of LLM on imperfect information games. The authors show superior performance of their agent w.r.t. traditional algorithms (such as CFR).

**Strengths:**

The study of LLM in decision making problem is currently one of the most important topic of research. This paper studies the applicability of LLM in imperfect information games and uses theory of main planning to infer private information based on the observed history. Moreover the topics of the paper are in line with the interests of the ICLR community.

**Weaknesses:**

In the reviewer's opinion the evaluation of the method is conceptually flawed. First, Nash equilibrium is never mentioned in the paper, which is the correct way of assessing the performance of algorithms in games. Second, the opponents considered are not optimal choices, new algorithms such as CFR+ or its optimistic variants (e.g. [1,2]) are now the state of the art in solving such games. DQN, DMC and NFSP are deep learning approaches that are not well motivated in such small games.
The main objective is the fallowing, any good algorithm in such games reaches ~0 exploitability (meaning that it finds optimal strategies, in terms of NE) and, be definition, no strategy can win against such strategies  when considering enough games. Thus either the CFR implementation is wrong (or far from convergence) or the evaluation methodology is flawed in other ways.

Moreover ToM s a valid approach only against exploitable agents, as the correct strategy against an agent with ~0 exploitability is itself a strategy with ~0 exploitability.

[1] Farina, Gabriele, Christian Kroer, and Tuomas Sandholm. "Optimistic regret minimization for extensive-form games via dilated distance-generating functions." Advances in neural information processing systems 32 (2019).
[2] Farina, Gabriele, et al. "Stable-predictive optimistic counterfactual regret minimization." International conference on machine learning. PMLR, 2019.

**Questions:**

1) Do you consider only two players games?
2) Way do you make a distinction between 0 and 1 position?
3) How is it possibile to consistently win against a strategy that computes NE?
4) Way are the evaluation made in terms of win/loss instead of exploitability as common in GT?

---

> ### Author Response · Authors · 2023-11-23
> **Response to  Reviewer iJUa**
>
> Thank you for your time and effort in sharing critical feedback regarding our work. We provide the following response to address your concerns about results limitations, and thus respectfully hope you can consider the response to the final decision.
>
> 1. the evaluation of the method is conceptually flawed.
>
> A: We acknowledge the concerns you raised regarding the experiments in our paper, and the omission of Nash equilibrium, a pivotal concept in evaluating algorithms in game settings. To address this, we added the relevant discussion in our paper, added the concept to the main paper and conducted additional experiments, specifically focusing on the CFR+ algorithm. Given the codebase of RLCard (https://github.com/datamllab/rlcard), we further train the CFR+ algorithm for 2000000 iterations and play our suspicion agent with CFR+, getting the following results:
> |                      |     CFR+ |  Ours  (GPT-3.5) |  Ours (GPT-4) |
> |:--------------------:|:--------------------------------------------------------:|:------------------------------------------------------:|:------------------:|
> |    CFR+      |  - |                     +126                   |                      +22                      |
> |      Ours (GPT-3.5)  | -126 |                             -                            |
> |     Ours (GPT-4)     |   -22  |                             -                            |                            -                           |
>
> which indicates the current LLM (without training examples and fine-tuning) cannot beat algorithms that reach Nash equilibrium when the iterations are large enough. Therefore, we have moderated our claims in the paper. Our revised findings demonstrate that our LLM may not surpass Nash Equilibrium strategies. **As a preliminary experimental study, Our focus is not on achieving Nash Equilibrium directly with the LLM but on exploring the design of an effective prompting system that can unlock the potential of LLMs in these complex game environments.** Our results highlight that  Suspicion-Agent can adaptively alter its behavior patterns when facing different opponents. This ability to effectively compete against simpler learning-based methods, which utilize reinforcement learning, is a noteworthy contribution to the field. Our research provides a foundation for further exploration into how LLMs can be utilized and improved for strategic decision-making in imperfect information games.
>
> 2. Do you consider only two players games?
>
> Yes, as a preliminary proof-concept study, we only consider two players games to make the problem simplified, and we would like extend it into multi-agent (more than 2) in the subsequent works.
>
> 3. Why do you make a distinction between 0 and 1 position?
>
> a. motivation of making a distinction between 0 and 1 position?
>
> Due to budget constraints, we were unable to conduct large-scale experiments like previous methodologies which involved over 100,000 games.  (**However, the distributions of strong hand and weak hands are not equal for each player in limited game rounds**)To ensure a fair comparison, it was crucial that both the Suspicion-Agent and the opponent model experienced the same card strength across an equal number of games. To achieve this balance, we employed a strategic approach: initially, the Suspicion-Agent was placed in position 0 for the first set of 50 games. Subsequently, we re-conduct these games but switched the positions, placing the Suspicion-Agent in position 1 and the baseline model in position 0. This method allowed us to maintain an equal card strength for both the Suspicion-Agent and the baseline model over a total of 100 games, thereby enabling a more accurate evaluation of each model's performance.
>
> b. Way to make it:
>
> To ensure consistency and reproducibility in our experiments, we used the same seed to establish the gaming environments. This approach allowed us to replicate the card distributions in different runs, facilitating the testing of the Suspicion-Agent in varying positions.
>
> 4. Additional optimistic variants:
>
> A: Thanks for your suggestions, we add the discussion about these works to the updated version.

---

> > ### Author Response · Authors · 2023-11-23
> > **Response2 to Reviewer iJUa**
> >
> > **In addition to the new results, as the first work to design the prompting system to enable LLM for imperfect information games, we still believe our proposed methods and the preliminary public experimental study and data are valuable for the community.**
> >
> > 1. We experimentally demonstrate that even with pre-trained data, **LLM itself  cannot play various imperfect information games well as Table 3 shows** ((-72 no ToM vs +24 second-order ToM)). Then we introduce the **first prompting system** to enable large language models like GPT-4 to play various imperfect information games using only the game rules and observations without any extra training, this may  inspire more LLM-based subsequent works.
> > 2. Leveraging the theory-of-mind capabilities of large language models, our work is **the first to demonstrate that a simple prompting system can enable GPT-4 to outperform some learning-based algorithms like DMC and NFSP, even without specialized training or examples. However, the further experiments prove that it still cannot beat algorithms that can achieve** Nash-Equilibrium such as CFR+. In addition, we also qualitatively evaluate the potential of LLM into the different imperfect information games,  which may inspire more subsequent works.
> > 3. We comprehensively identify and discuss the current limitations of employing large language models in imperfect information games, contributing valuable insights for the further AI-Agent research.
> > 4. We are preparing to make all our code and interactive data publicly available. We hope that this will advance the community's understanding of the capabilities of large language models, particularly GPT-4. We also hope this will catalyze the development of more efficient models in the field of imperfect information games.

---

### Author Response · Authors · 2023-11-23
**General Response**

Thanks for the time and effort sharing critical feedback of every reviewer regarding our work. In order to address the questions and points raised by the reviewers, we have provided additional experimental results in our response PDF.

1. Following the suggestions of reviewers, we update the results with CFR+, and find that current prompting design cannot beat algorithm reach Nash equilibrium. However, **as a preliminary experimental study, our primary objective is not to directly achieve Nash Equilibrium with the LLM but to explore the development of an effective prompting system. This system aims to harness the potential of LLMs in complex game environments and to identify their current limitations.** Our results highlight that Suspicion-Agent can adaptively alter its behavior patterns when facing different opponents. This ability to effectively compete against simpler learning-based methods, which utilize reinforcement learning, is a noteworthy contribution to the field. Our research provides a foundation for further exploration into how LLMs can be utilized and improved for strategic decision-making in imperfect information games.

2.  To improve the randomness of our experiments, we already designed two types of experiments to compare Suspicion Agent with baselines:

a. Reducing the randomness of Random Seeds:

Our agent plays against different baselines for 100 games utilizing varying random seeds for each game. This tactic is intended to dampen the stochastic variability introduced by the random seed settings.

b. Reducing the randomness of Card Strength/Position: (which is also used in the recommended AIVAT [1])

Given the limited games, the distributions of strong hand and weak hands are not equal for each player in limited game rounds. To ensure a fair comparison, it was crucial that both the Suspicion-Agent and the opponent model experienced the same card strength across an equal number of games. To achieve this balance, we employed a strategic approach: initially, the Suspicion-Agent was placed in position 0 for the first set of 50 games. Subsequently, we reproduce these 50 games with same cards as the first one but switched the positions, placing the Suspicion-Agent in position 1 and the baseline model in position 0. This method allowed us to maintain an equal card strength for both the Suspicion-Agent and the baseline model over a total of 100 games, thereby enabling a more accurate evaluation of each model's performance.

Given these two designs, we believe that we can alleviate the effect of randomness in imperfect information games. Given the results in Table 1 and Table2 in the main paper: Suspicion Agent achieves superior performance over learning-based baselines except CFR+ in all 100 random games, 50 games in position 0 and 50 games in position 1. They are the clear evidence to show the advantages of our suspicion agent.

3. **In addition to the new results, as the first work to design the prompting system to enable LLM for imperfect information games, we still believe our proposed methods and the preliminary public experimental study and data are valuable for the community.**

a. We experimentally demonstrate that even with pre-trained data, **LLM itself  cannot play various imperfect information games well as Table 3 shows** ((-72 no ToM vs +24 second-order ToM)). Then we introduce the **first prompting system** to enable large language models like GPT-4 to play various imperfect information games using only the game rules and observations without any extra training, this may  inspire more LLM-based subsequent works.

b. Leveraging the theory-of-mind capabilities of large language models, our work is **the first to demonstrate that a simple prompting system can enable GPT-4 to outperform some learning-based algorithms like DMC and NFSP, even without specialized training or examples. However, the further experiments prove that it still cannot beat algorithms that can achieve** Nash-Equilibrium such as CFR+. In addition, we also qualitatively evaluate the potential of LLM into the different imperfect information games,  which may inspire more subsequent works.

c. We comprehensively identify and discuss the current limitations of employing large language models in imperfect information games, contributing valuable insights for the further AI-Agent research.

d. We are preparing to make all our code and interactive data publicly available. We hope that this will advance the community's understanding of the capabilities of large language models, particularly GPT-4. We also hope this will catalyze the development of more efficient models in the field of imperfect information games.

Considering these contributions, we respectfully hope that the reviewers take our comprehensive response and the data provided into account when making their final decision.

[1] Burch, Neil, et al. "Aivat: A new variance reduction technique for agent evaluation in imperfect information games." AAAI  2018.

---

### Meta-Review · Area_Chair_gZPv · 2023-12-06

**Metareview:**

Here is a summary of the remaining concerns after author responses:

> The present experimental results may not be sufficiently reliable because of limited number of game runs.
- We recommend that the authors refer to prior work for a more comprehensive assessment of the performance of incomplete information agents.

> Limited Evaluation Games and Generalizability (quantitative evaluation on a single game and qualitative evaluation on three games).
- The limited evaluation games makes it difficult to estimate the generalizability of the proposed approach. Moreover, in the related work especially published in the NLP community, a diverse set of different ToM dimensions are assessed. The proposed work covers a special aspect of ToM so its true value is not clear.

> The game may have the winning strategy seen during LLM training. It is unclear whether the proposed prompting approach helps the LLMs recall the memory of the winning strategies.
- This issue is recognized as one important challenge to deal with when assessing LLMs' ToM abilities, since the first tech report of GPT4. To address this, it is also advisable to consider designing new games with similar settings but adapted descriptions or rules, as recommended by peer-reviewed publications in the ToM-related areas.

> Unclear performance against human players.
- Although the authors have conducted new experiments, the details regarding their performance against human players remain unclear. More transparent reporting of these experiments would be beneficial to provide a better understanding of the model's capabilities.

> (Kind of) overclaim on "first work to design the prompting system to enable LLM for imperfect information games".
- Considering of a broader perspective of imperfect information games, there have been prior work on LLMs for this purpose. For example, Werewolf can also be regarded as an imperfect information game

**Justification For Why Not Higher Score:**

- The game may have the winning strategy seen during LLM training. It is unclear whether the proposed prompting approach helps the LLMs recall the memory of the winning strategies.
- The present experimental results may not be sufficiently reliable because of limited number of game runs.
- Limited Evaluation Games and Generalizability (quantitative evaluation on a single game and qualitative evaluation on three games).
- Unclear performance against human players.
- (Kind of) overclaim on "first work to design the prompting system to enable LLM for imperfect information games"

**Justification For Why Not Lower Score:**

N/A - rejection

---

### Decision · Program_Chairs · 2024-01-16

Reject